# Adaptive Time Series Reasoning via Segment Selection

**Shvat Messica** [1]   **Jiawen Zhang** [* 2]   **Kevin Li** [* 3 1]   **Theodoros Tsiligkaridis** [4]   **Marinka Zitnik** [1]

## Abstract

Time series reasoning tasks often start with a natural language question and require targeted analysis of a time series. Evidence may span the full series or appear in a few short intervals, so the model must decide what to inspect. Most existing approaches encode the entire time series into a fixed representation before inference, regardless of relevance. We introduce ARTIST, which formulates time-series reasoning as a sequential decision problem. ARTIST interleaves reasoning with adaptive temporal segment selection. It adopts a controller-reasoner architecture and uses reinforcement learning to train the controller role to select informative segments and the reasoner role to generate segment-conditioned reasoning traces and final answers. During inference, the model actively acquires task-relevant information instead of relying on a static summary of the full sequence. We use a novel hierarchical policy optimization approach for post-training that allows the model to excel in both segment selection and question-answering behavior. We evaluate ARTIST on six time-series reasoning benchmarks and compare it with large language models, vision-language models, and prior time-series reasoning systems. ARTIST improves average accuracy by 6.46 absolute percentage points over the strongest baseline. The largest gains appear on rare event localization and multi-segment reasoning tasks. Supervised fine-tuning improves performance, and reinforcement learning provides additional gains by optimizing question-adaptive segment selection, showing that selective data use drives effective time-series reasoning.

*Equal contribution [1]Department of Biomedical Informatics, Harvard Medical School, Boston, MA, USA [2]The Hong Kong University of Science and Technology (Guangzhou) [3]Massachusetts Institute of Technology [4]MIT Lincoln Laboratory. Correspondence to: Shvat Messica and Marinka Zitnik <shvat.messica@fas.harvard.edu, marinka@hms.harvard.edu>.

*Proceedings of the 43rd International Conference on Machine Learning*, Seoul, South Korea. PMLR 306, 2026. Copyright 2026 by the author(s).

## 1. Introduction

Time-series modeling has long been shaped by tasks such as forecasting (Das et al., 2024; He et al., 2025a; Yang et al., 2025b; Wang et al., 2025b), classification (Liu et al., 2025d;c; Tran et al., 2026), and anomaly detection (Sun et al., 2026; Cai et al., 2026; Kim et al., 2025). Increasingly, however, real-world use cases begin with a natural-language question rather than a predefined label or prediction horizon. A clinician may ask why a patient's vital signs deteriorated after a medication change, and an analyst may ask what triggered a regime shift in a market indicator. Answering these questions requires linking the question to patterns in the time series and grounding the answer in relevant segments. This emerging setting, known as time series reasoning (Chow et al., 2024b; Kong et al., 2025b), asks models to answer natural language questions by identifying and analyzing the task-relevant parts of a time series.

Recent work has examined how to interface time series with large language models (LLMs). As in Figure 1a, approaches mainly differ in how they convert a time series into an LLM input, including textual serialization (Al-negheimish et al., 2024; Luo et al., 2025; Liu et al., 2025e), rendered plots (Liu et al., 2025b; Zhang et al., 2025a), and learned embeddings (Xie et al., 2024; Lei et al., 2025; Langer et al., 2025). These designs improve modality alignment but keep inference static: the model receives a fixed view of the full time series and must answer from that view. This is limiting for multi-step reasoning, where intermediate conclusions should change what the model inspects next. For example, a clinical deterioration query may first check for an abrupt spike, then shift to a different window or a narrower interval to test a hypothesis. More broadly, relevant temporal segments are query- and step-dependent. Figure 1b shows the desired behavior: reasoning interleaved with temporal segment selection, stopping when the model is ready to answer.

A key missing capability is question-adaptive information selection during inference. Time series inputs are long and temporally heterogeneous, and the same sequence can support many different questions that require different regions and different resolutions. As a result, forcing the model to process a fixed view of the full series can dilute salient local structure and, in long sequences, can degrade performance

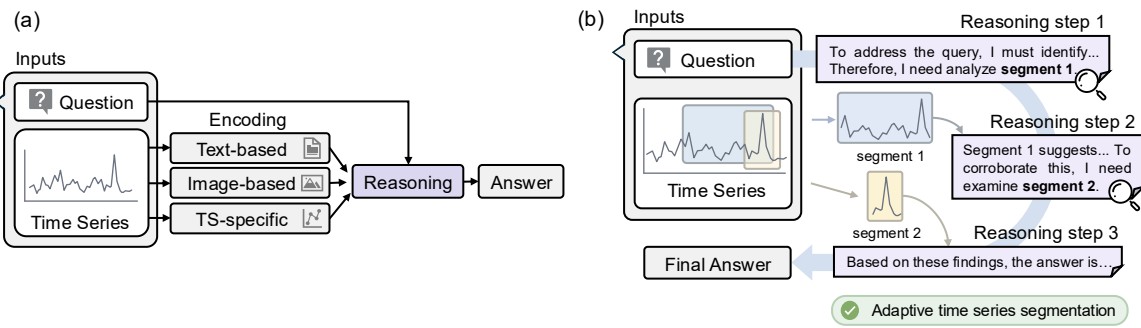

*Figure 1.* (a) Time-series reasoning: answering a natural language question given a time series. (b) ARTIST alternates between reasoning and adaptive segment selection, choosing the next segment based on the question and intermediate outputs, and stopping once it can produce the final answer.

by mixing irrelevant context with task-relevant information. Related problems have been studied in text-vision reasoning, where models learn to search or zoom into regions conditioned on intermediate steps (Su et al., 2026; Zhang et al., 2025c; Li et al., 2025). However, an analogous mechanism for time-series reasoning remains underexplored, despite the strong need for adaptive localization in long time series.

Two challenges make question-adaptive segment selection difficult in time series reasoning. First, unlike images, time series rarely include task-agnostic annotations that identify segments relevant to a given question. Relevant segments depend on the question, so the model must decide during inference which parts of time series to inspect. Second, many post-training methods use reinforcement learning to optimize sequential decision making at the level of output tokens (e.g., PPO (Schulman et al., 2017), GRPO (Shao et al., 2024), DAPO (Yu et al., 2025a)). Token-level optimization becomes difficult when a task requires multiple sub-tasks and the reasoning horizon is long. Credit assignment then becomes diffuse: key steps such as tool calls or information selection occupy only a small part of the token sequence, so their effect on final output is hard to isolate and optimize. Approaches that disentangle traces into components and optimize them with role-specific objectives can target these steps directly (Klissarov et al., 2025). This motivates our present work on separating temporal segment acquisition from time series answer generation.

**Present work.** ARTIST[1] (Adaptive Reasoning for TIme-Series via Temporal Selection) is a time-series reasoning approach where an LLM is trained to treat time series as an active resource during inference (Figure 1). It interleaves reasoning with temporal segment selection by training a single policy to operate in two roles. A high-level *controller* selects the next segment and decides when to stop, conditioned on the question and intermediate outputs. A

low-level *reasoner* produces intermediate reasoning steps and the final answer conditioned on the selected segments. This separation explicitly decouples *where to look* from *how to reason*. ARTIST is trained in two stages. First, we apply supervised fine-tuning (SFT) on curated traces that interleave segment selection with natural language reasoning. Second, we use collaborative self-play reinforcement learning to jointly optimize both roles. The controller is trained with a reliability-based reward that measures answer consistency under repeated sampling, and the reasoner is trained for correctness and format compliance conditioned on controller-selected segments. Across six benchmarks in clinical, financial, environmental, and general domain time series reasoning, ARTIST outperforms large language models, vision-language models, and prior time-series reasoning systems. It achieves an average accuracy improvement of 6.46 percentage points over the strongest comparator, with gains of up to 12.5 points on datasets requiring localized or multi-segment reasoning. Supervised fine-tuning alone yields an average accuracy increase of 5.65 points over baselines, while reinforcement learning provides additional gains by optimizing question-adaptive segment selection. Further, ARTIST typically uses only 30-70% of the input time series sample per query, showing that selective temporal analysis improves accuracy without requiring full time series consumption.

**Contributions.** (1) We present adaptive segment selection for time series reasoning. The model iteratively chooses temporal segments to inspect and updates its reasoning based on the retrieved segments. This setting does not assume pre-defined segment labels. (2) We introduce a hierarchical, collaborative self-play RL post-training method that separates segment selection from answer generation and trains each role with learning signals aligned to its role. (3) On six benchmarks, ARTIST outperforms seven strong baselines. It improves average accuracy by 6.46 absolute percentage points over the strongest comparator while using a smaller fraction of the input time series.

**Conflict of Interest Disclosure.** The authors declare no

---

[1]ARTIST website: https://zitniklab.hms.harvard.edu/ARTIST/; code: https://github.com/mims-harvard/ARTIST

financial conflicts of interest or other substantive conflicts that could reasonably be perceived to influence the work presented in this paper.

## 2. Related Work

**Time Series Reasoning.** Motivated by the need to solve real-world questions that require reasoning over evolving signals, as well as by advances in multimodal LLMs capable of processing non-textual inputs, recent studies have begun to formalize time-series reasoning. Several benchmarks have been introduced to evaluate such capabilities by pairing time-series signals with natural-language queries, including ECG-QA (Oh et al., 2023), TSQA (Kong et al., 2025a), TRQA (Jing et al.), RCW (Liu et al., 2025b), and related datasets (Merrill et al., 2024). These resources frame temporal reasoning as question answering, explanation, or diagnostic inference over real or synthetic data, providing standardized settings for assessing LLM-based approaches. In parallel with benchmark development, several modeling strategies have emerged to support time-series reasoning in LLMs. One line of work treats time-series values as textual inputs, allowing standard text-only LLMs to process temporal data without architectural modification. A second line couples a time-series encoder with a language model, such as ChatTS (Xie et al., 2024), OpenTSLM (Langer et al., 2025), and ITFormer (Lei et al., 2025), mapping numerical observations into learned embeddings that the LLM can condition on for reasoning. A third line of work encodes time series into visual representations. VL-Time (Liu et al., 2025b) and TimeMaster (Zhang et al., 2025a) render signals as plots and use vision-language models to interpret them. These methods improve performance on time-series reasoning tasks, but they mainly address representation and modality integration, that is, how temporal signals are encoded and adapted for LLM processing. In contrast, the problem of selecting and analyzing temporal information during reasoning remains underexplored.

**Dynamic Visual Search in Vision-Language Models.** A related line of work in vision-language reasoning focuses on developing models that iteratively select or zoom in on informative image regions conditioned on intermediate reasoning steps, acquiring additional visual information as needed to solve a task. Prompt-based methods such as ZoomEye (Shen et al., 2025) and Chain-of Spot (Liu et al., 2024a) use tree-based search or question-conditioned region-of-interest cropping to localize question-relevant regions, while Set-of-Mark prompting (Yang et al., 2023) and DetToolChain (Wu et al., 2024) leverage segmentation-based visual marks or detection-oriented prompting toolkits to ground reasoning in specific regions. More recent reinforcement learning approaches extend this paradigm

with learned selection policies: Chain-of-Focus (Zhang et al., 2025c) trains a model to progressively zoom into regions based on intermediate cues, DeepEyes (Zheng et al., 2026) interleaves multimodal chain-of-thought with native visual grounding, and Pixel-Reasoner (Su et al., 2026) uses curiosity-driven RL to incentivize pixel-space exploration.

While these methods share ARTIST's core principle of treating perceptual input as an active resource to be iteratively queried rather than as static context, time series reasoning differs from visual search in both problem setting and learning signal. Vision tasks typically target entities with well-defined spatial boundaries and often benefit from explicit supervision such as bounding boxes or segmentation masks, whereas temporal segments lack task-agnostic annotations and their relevance is inherently question-dependent, requiring ARTIST to learn selection through weak supervision from downstream reasoning outcomes. Segments are also not self-contained: a zoomed image region is often interpretable in isolation, whereas a temporal segment's meaning depends on other segments - a spike is informative only relative to a baseline, and a regime shift requires comparing intervals before and after, so selection must be sequential and context-aware. ARTIST's controller-reasoner decomposition and reliability-based reward target these challenges directly.

**Self-Play RL for Reasoning.** Self-play originated in game theory and reinforcement learning as a mechanism for learning in adversarial multi-agent settings, where agents improve by interacting with copies of themselves or past versions of their policies rather than with a fixed opponent, and where outcomes depend directly on the evolving strategies of other agents, requiring continual adaptation instead of optimization against a static objective (Zhang et al., 2025b). More recently, this paradigm has been adapted to LLMs, where self-play is used to generate training signals and improve reasoning without relying on human-labeled data (Zhao et al., 2025; Fang et al., 2025; Liu et al., 2025a; Yang et al., 2025c; Huang et al., 2025; Chen et al., 2025; He et al., 2025b; Wang et al., 2025a; Yu et al., 2025b; Yue et al., 2026; Wang et al., 2026). In these studies, a single LLM assumes multiple roles within the same training loop, typically forming a difficulty-driven interaction. A *proposer* generates tasks that challenge the model's current capabilities, while a *solver* attempts to solve them and is rewarded based on correctness. This alternation exploits the stateless nature of LLMs, allowing the same underlying model to perform distinct roles via prompting, thereby enabling data- and memory-efficient training without maintaining multiple separate models. As a result, self-play provides a mechanism for refining model behavior through role-based interaction and outcome evaluation, rather than reliance on external human annotations. This is important for time-series reasoning and adaptive segment selection,

where annotations specifying which temporal segments are relevant to a given question are difficult to obtain.

In contrast to adversarial self-play, ARTIST uses collaborative self-play to separate time-series reasoning into distinct roles. ARTIST assigns adaptive segment selection to a controller and answer construction to a reasoner. The controller selects temporal segments. The reasoner produces intermediate reasoning steps and the final answer conditioned on the selected segments. This separation distinguishes evidence acquisition from answer generation and supports role-specific learning signals. Existing self-play methods, including AZR (Zhao et al., 2025), SPICE (Liu et al., 2025a), and SPELL (Yang et al., 2025c), optimize each role with myopic objectives that reward the best immediate response within a single round. These objectives do not match temporal segment selection, where the model must build an informative set of segments across multiple rounds. ARTIST addresses this mismatch with hierarchical optimization. The controller receives trajectory-level supervision with credit assigned across interaction rounds, while the reasoner is optimized at the final round conditioned on the selected segments.

# 3. ARTIST

## 3.1. Problem Setup and Notation

We study *time-series reasoning* for a natural-language question $q$ paired with a time series $T \in \mathbb{R}^{H \times V}$, where $H$ is the sequence length and $V$ is the number of variables. A time-series large language model $\pi_\theta$ generates a textual response $Y$:

$$\pi_\theta(q, T) \to Y. \tag{1}$$

In this paper, we focus on the univariate setting ($V = 1$).

**Time Series Reasoning.** We view reasoning as searching for a high-probability inference path that links $(q, T)$ to a correct answer. Concretely, we introduce a latent sequence of intermediate steps $z = (z_1, \dots, z_L)$ and write:

$$p_\theta(Y \mid q, T) = \sum_{z \in \mathcal{Z}} p_\theta(Y \mid z, q) \, p_\theta(z \mid q, T), \tag{2}$$

where $z$ specifies *how* the model inspects and interprets the signal before producing $Y$.

**Segments.** A segment is a contiguous slice of the time series $T$, denoted $s = T_{t_{\text{start}}:t_{\text{end}}}$ with $0 \leq t_{\text{start}} < t_{\text{end}} \leq H$. We maintain an accumulated segment list $\mathcal{S} = [s^1, \dots, s^K]$ over multiple reasoning steps.

## 3.2. Controller-Reasoner Roles in ARTIST

ARTIST is a collaborative self-play model that equips a single policy model $\pi_\theta$ with both *adaptive segment local-ization* and *segment-conditioned reasoning*. The same underlying model is invoked in two roles via role-specific prompting, yielding *controller* $\pi_\theta^{\text{ctl}}$ and *reasoner* $\pi_\theta^{\text{rsn}}$. The controller decides which temporal segments to inspect and when to terminate the interaction, while the reasoner generates a reasoning trace and a final answer conditioned on the accumulated segments.

Each training iteration consists of (i) generating rollouts through Controller-Reasoner interaction and (ii) computing a joint policy update. Figure 2 illustrates the rollout and update structure, and Algorithm 1 summarizes the optimization process.

## 3.3. Interaction Trajectory Rollout

For a given question-time series pair $(q, T)$, an *interaction trajectory* $\tau$ is a sequence of Controller-Reasoner interaction rounds indexed by $i = 1, 2, \dots, L$, where $L$ is the realized number of rounds until termination.

**Controller Step.** At round $i$, the Controller receives the current state:

$$x_i^{\text{ctl}} = (q, T, \mathcal{S}_{i-1}, a_{i-1}, \hat{y}_{i-1}), \tag{3}$$

where $(\mathcal{S}_0, a_0, \hat{y}_0)$ are empty at initialization. In all subsequent interaction rounds $i > 1$, $a_{i-1}$ and $\hat{y}_{i-1}$ are the reasoning trace and answer provided by the Reasoner in the previous round $i - 1$. The controller then outputs a reasoning trace $u_i$ and a termination decision $d_i \in \{\text{CONTINUE}, \text{ACCEPT}\}$. If $d_i = \text{CONTINUE}$, it additionally proposes a new segment $s_i$:

$$(u_i, d_i, s_i) \sim \pi_\theta^{\text{ctl}}(\cdot \mid x_i^{\text{ctl}}), \tag{4}$$

and we append the new segment to the segment list $\mathcal{S}_{(i)} \leftarrow \mathcal{S}_{i-1} \cup \{s_i\}$. If $d_i = \text{ACCEPT}$, the interaction terminates and the answer outputted by the Reasoner step in the previous round, $\hat{y}_{i-1}$, is accepted as the final answer.

**Reasoner Step.** If $d_i = \text{CONTINUE}$, the Reasoner is invoked with the updated segment list $\mathcal{S}_i$ and produces a reasoning trace $a_i$ and answer $\hat{y}_i$:

$$(a_i, \hat{y}_i) \sim \pi_\theta^{\text{rsn}}(\cdot \mid q, \mathcal{S}_i). \tag{5}$$

Formally, we define an interaction trajectory as:

$$\tau = \big\{(u_i, s_i, d_i, a_i, \hat{y}_i)\big\}_{i=1}^{L}, \qquad d_L = \text{ACCEPT}. \tag{6}$$

We further define two sub-trajectories of an interaction trajectory: *controller trajectory* as $\tau_{\text{ctl}} = \{(u_i, s_i, d_i)\}_{i=1}^{L}$, and the *reasoner trajectory @ round i* as $\tau_{\text{rsn, i}} = (a_i, \hat{y}_i)$.

## 3.4. Reward Functions

We define rewards for the two roles based on answer correctness, reliability across repeated trials, and format compliance.

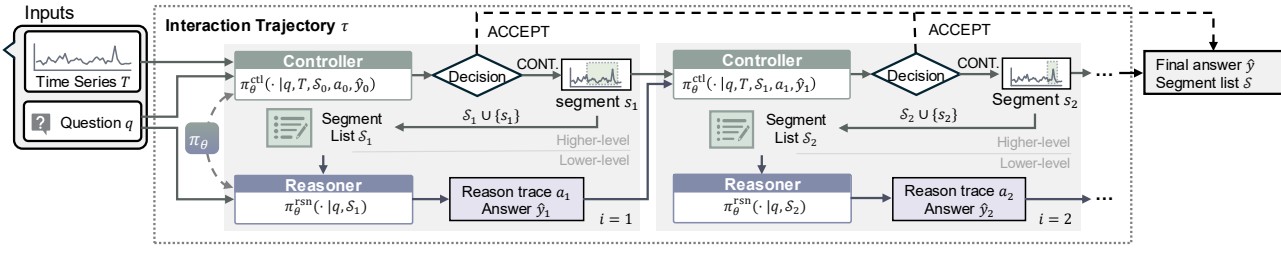

*(a)* Interaction Trajectory Rollout (Generation).

*(b)* Unified Policy Update (Optimization).

*Figure 2.* **Overview of ARTIST.** (a) Given a question and a time series, a high-level controller iteratively selects informative temporal segments, while a low-level reasoner produces intermediate reasoning traces and answers conditioned on the accumulated segment set. The controller decides whether to continue selecting or accept it as a final answer and segment list. (b) For each training example, multiple reasoning rollouts are generated per controller trajectory. Reasoner rewards are group-normalized across reasoning rollouts, while controller rewards are computed using the associated reasoning outcomes and normalized across controller trajectories. Both signals are used for a joint policy update.

**Correctness Reward.** Given a question $q$, gold answer $y^*$, and prediction $\hat{y}$, we define the correctness reward as:

$$C(q, y^*, \hat{y}) = \mathbf{1}[\hat{y} = y^*]. \tag{7}$$

**Reliability Reward.** Given $q$ and a segment list $\mathcal{S}$, we define reliability as the expected correctness of the Reasoner under repeated sampling conditioned on $(q, \mathcal{S})$. In practice, we estimate it with $N$ independent rollouts $\hat{y}^{(n)} \sim \pi_\theta^{\text{rsn}}(\cdot \mid q, \mathcal{S}), n = 1, \ldots, N$ and then calculate:

$$
\begin{aligned}
D(q, \mathcal{S}, y^*) &= \frac{1}{N} \sum_{n=1}^{N} \mathbf{1}\left[\hat{y}^{(n)} = y^*\right] \\
&= \frac{1}{N} \sum_{n=1}^{N} C(q, y^*, \hat{y}^{(n)}).
\end{aligned} \tag{8}
$$

A higher reliability reward indicates that the selected segments $\mathcal{S}$ make the Reasoner consistently produce the correct answer under sampling noise, and we use it as the learning signal for the Controller.

**Controller Format Score.** Each controller step receives a per-step format score $f_i$ that captures minor but executable errors such as multiple reasoning blocks (see Appendix A.6). Given controller trajectory $\tau_{\text{ctl}} = \{(u^i, d^i, s^i)\}$, we additionally define a critical violation indicator $\mathbb{I}_{\text{viol}}(\tau_{\text{ctl}}) \in \{0, 1\}$ that flags non-executable outputs (full list in Appendix A.6). We then define:

$$F_{\text{ctl}}(\tau_{\text{ctl}}) = \left(1 - \mathbb{I}_{\text{viol}}(\tau_{\text{ctl}})\right) \cdot \frac{1}{L} \sum_{i=1}^{L} f_i - \mathbb{I}_{\text{viol}}(\tau_{\text{ctl}}). \tag{9}$$

**Controller Reward.** The controller reward is defined as:

$$R_{\text{ctl}}(\tau_{\text{ctl}}, D) = \begin{cases} -1 & \text{if } F_{\text{ctl}}(\tau_{\text{ctl}}) < 0, \\ w_D D + w_f F_{\text{ctl}}(\tau_{\text{ctl}}) & \text{otherwise,} \end{cases} \tag{10}$$

where $w_D, w_f$ are tunable hyperparameters.

**Reasoner Reward.** Given a final-round reasoner trajectory $\tau_{\text{rsn}} = (a, \hat{y})$ with correctness $c = C(q, y^*, \hat{y})$,

$$R_{\text{rsn}}(\tau_{\text{rsn}}, c) = w_c\, c + w_e\, F_{\text{rsn}}(\tau_{\text{rsn}}), \tag{11}$$

where $F_{\text{rsn}}$ scores format compliance (Appendix A.6) and $w_c, w_e$ are tunable hyperparameters.

### 3.5. Hierarchical Policy Optimization

As shown in Fig. 2b, we use nested rollouts to obtain separate learning signals for the Controller and Reasoner behavior, and combine them into a joint update. Decomposing the parameter update in this way enables more precise credit assignment, as advantage signals are propagated only to the relevant tokens for each role.

**Nested Rollouts.** For each training example $(q, T, y^*)$, we first sample $G$ interaction trajectories $\{\tau^{(g)}\}_{g=1}^{G}$ following Sec. 3.3. Let the terminating round of $\tau^{(g)}$ be $L^{(g)}$, and let the final segment list be $\mathcal{S}_{L^{(g)}}^{(g)}$ (ACCEPT does not add a new segment). For each interaction trajectory $g$, we then resample the Reasoner $N$ times conditioned on the same final segment list, and denote these final-round reasoner trajectories as $\tau_{\text{rsn}, L^{(g)}}^{(g,n)} = (a_{L^{(g)}}^{(g,n)}, \hat{y}_{L^{(g)}}^{(g,n)}), n = 1, \ldots, N$.

We then compute the mean and variance correctness re-

wards for reasoner trajectories, i.e., $r_\mu^{(g)}$ and $r_\sigma^{(g)}$, where $r_\mu^{(g)}$ equals the reliability reward in Eq. (8) for the final segments $\mathcal{S}_{L^{(g)}}^{(g)}$ produced in rollout $g$.

**Higher-Level Advantage Calculation for Controller.** The Controller is optimized as a high-level policy that iteratively localizes segments across multiple interactions with the Reasoner and decides when the current information is sufficient. For each interaction rollout $g \in \{1, \dots, G\}$, we compute a Controller reward $r_{\text{ctl}}^{(g)} = R_{\text{ctl}}(\tau_{\text{ctl}}^{(g)}, r_\mu^{(g)})$ for controller trajectory $\tau_{\text{ctl}}^{(g)}$. We then form group-relative advantages $\{\hat{A}_{\text{ctl}}^{(g)}\}_{g=1}^G$ according to Appendix A.1.

**Lower-Level Advantage Calculation for Reasoner.** The Reasoner is optimized as a low-level policy that maximizes answer quality and formatting conditioned on the segments selected by the Controller. Unlike the Controller, the Reasoner update optimizes only the final-round rollouts under a fixed segment list, disentangling the optimization of Reasoner behavior from the variance in quality of the segments selected by the Controller. For each interaction rollout $g$, we draw $N$ independent final-round Reasoner rollouts $\{\tau_{\text{rsn},L^{(g)}}^{(g,n)}\}_{n=1}^N$ under the same $\mathcal{S}$.

Using all $G \times N$ rollouts for every update can be memory-intensive, and many groups provide limited learning signal when their rewards are nearly identical. We therefore update the Reasoner using a single group $g^*$ selected by *variance-guided sampling*, which prioritizes interaction trajectory that induce diverse outcomes and thus yield a more informative within-group advantage signal:

$$p(g) \; \propto \; r_\sigma^{(g)}, \qquad (12)$$

where $r_\sigma^{(g)}$ is the variance of correctness across the $N$ rollouts in group $g$. For the sampled group $g^*$, we compute a Reasoner reward for each trajectory $\tau_{\text{rsn}}^{(g^*,n)}$:

$$r_{\text{rsn}}^{(g^*,n)} \; = \; R_{\text{rsn}}\Big(\tau_{\text{rsn},L^{(g)}}^{(g^*,n)}, \, C\big(q, y^*, \hat{y}_{L^{(g)}}^{(g^*,n)}\big)\Big), \qquad (13)$$

and form group-relative advantages $\{\hat{A}_{\text{rsn}}^{(g^*,n)}\}_{n=1}^N$ according to Appendix A.3.

**Joint Update.** Finally, we aggregate the Controller advantages $\{\hat{A}_{\text{ctl}}^{(g)}\}_{g=1}^G$ and the Reasoner advantages $\{\hat{A}_{\text{rsn}}^{(g^*,n)}\}$ into a joint update of the shared parameters $\theta$ of $\pi_\theta$ that maximizes the objectives in Appendix A. Crucially, our Controller advantage signals are propagated across the Controller outputs from *all* rounds $i \in \{1, \dots, L^{(g)}\}$ in each interaction trajectory $\tau^{(g)}$, optimizing the role's behavior for incrementally choosing segments over multiple interactions that eventually result in the best segment *set* long-term. In contrast, the Reasoner advantage signals are propagated only to the Reasoner outputs from the last round $L^{(g)}$ in each $\tau^{(g)}$, optimizing the role's behavior myopically to encourage focused, granular reasoning during

question-answering. This hierarchical optimization setup is advantageous compared to previous self-play approaches for reasoning-based post-training that optimize both roles myopically, which would not be compatible for iterative segment selection.

---

**Algorithm 1** Hierarchical Policy Optimization

---
**Require:** initial policy model $\pi_{\theta_{\text{init}}}$, question $q$

  $\pi_\theta \leftarrow \pi_{\theta_{\text{init}}}$
  **for** $g = 1, 2, ..., G$ **do**
    $\tau^{(g)} \leftarrow$ interaction rollout of length $L^{(g)}$ using $\pi_\theta$
    **for** $n = 1, 2, ..., N$ **do**
      $\tau_{rsn,L^{(g)}}^{(g,n)} \leftarrow (a_{L^{(g)}}^{(g,n)}, \hat{y}_{L^{(g)}}^{(g,n)})$ (resampled reasoner rollout
      @ iter. $L^{(g)}$)
      $r_{\text{rsn}}^{(g,n)} \leftarrow R_{\text{rsn}}\Big(\tau_{\text{rsn}}^{(g,n)}, \, C\big(q, y, \hat{y}^{(g,n)}\big)\Big)$
    **end for**
    $r_\mu^{(g)} \leftarrow \text{Mean}(\{C\big(q, y, \hat{y}_{L^{(g)}}^{(g,n)}\big)\}_{n=1}^N)$
    $r_\sigma^{(g)} \leftarrow \text{Var}(\{C\big(q, y, \hat{y}_{L^{(g)}}^{(g,n)}\big)\}_{n=1}^N)$
    $r_{\text{ctl}}^{(g)} \leftarrow R_{\text{ctl}}\big(\tau_{\text{ctl}}^{(g)}, r_\mu^{(g)}\big)$
  **end for**
  Calculate advantages $\hat{A}_{\text{ctl}}^{(g)}$ using $\{r_\mu^{(g)}\}_{g=1}^G$
  Sample $g^* \sim \{1, 2, ..., G\} : p(g) \propto r_\sigma^{(g)}$
  Calculate advantages $\hat{A}_{\text{rsn}}^{(g^*,n)}$ using $\{r_{\text{rsn}}^{(g^*,n)}\}_{n=1}^N$
  Joint update using $\hat{A}_{\text{ctl}}^{(g)}$ and $\hat{A}_{\text{rsn}}^{(g^*,n)}$

---

## 4. Experiments

**Datasets.** We evaluate on six time-series reasoning benchmarks spanning diverse domains and question formats: TSQA (Kong et al., 2025a), RCW (Liu et al., 2025b), ECG-QA (Oh et al., 2023), Sleep-QA (Langer et al., 2025), TRQA (Jing et al.), and Etiological Reasoning (ETI) (Merrill et al., 2024). We follow each benchmark's standard evaluation protocol, applying only minor restrictions for consistency (details in Appendix B). Dataset statistics are summarized in Table 4.

**Baselines.** We compare against three families of models that support time-series reasoning. (1) Text-based LLMs include GPT-5 (Singh et al., 2025), LLaMA-3-8B (Grattafiori et al., 2024), Qwen-3-8B and Qwen-3-14B (Yang et al., 2025a), which take serialized time series as input. (2) Fine-tuned time-series reasoning models include ChatTS (Xie et al., 2024), OpenTSLM-Flamingo (Langer et al., 2025), and ITFormer (Lei et al., 2025), which pair a temporal encoder with an LLM backbone. (3) Vision-based baselines include Qwen2.5-VL-3B (Bai et al., 2025), VL-Time (Liu et al., 2025b), and TimeMaster (Zhang et al., 2025a), which operate on plotted time series. We use Qwen2.5-VL-3B-Instruct as backbone for vision-based models. For all fine-tuned baselines reported with a row of "+ SFT" in Table 1, supervised fine-tuning is performed on the same training data used for the

SFT stage of ARTIST.

**Implementation & Evaluation.** We demonstrate the ARTIST approach using a Qwen3-4B backbone and a 5-layer MLP that encodes patch-based time-series inputs. Segment selection is facilitated through a tool-calling mechanism that enables the model to iteratively retrieve temporal segments. Training proceeds in two stages: (1) SFT, utilizing LoRA-based fine-tuning on structured reasoning traces that interleave natural language with segment-selection calls (Appendix C); and (2) RL, employing full-parameter fine-tuning via our collaborative self-play algorithm. Performance is measured using average Accuracy and F1 scores across 8 independent runs per dataset. For ARTIST, we maintain fixed temperatures for the reasoner (0.7) and controller (1.0). See Appendix E for comprehensive hyperparameter settings.

### 4.1. Benchmarking Results

**Comparison with general LLMs and time-series reasoning models.** Table 1 shows that ARTIST achieves the best overall performance across all benchmarks. Relative to the strongest baseline on each dataset and metric, ARTIST improves average accuracy by 6.46%. RL further improves performance beyond SFT. Compared with SFT, adding RL increases average accuracy from 63.61% to 69.26%. **Comparison with vision–language reasoning models.** Table 2 compares ARTIST with VLM baselines that operate on plotted time series. ARTIST is best on five of six datasets (ETI, RCW, ECG-QA, TSQA, TRQA), while TimeMaster+RL performs better on Sleep-QA. We also observe that on ECG- and EEG-based datasets (ECG-QA and Sleep-QA), even VLMs without task-specific fine-tuning can be highly competitive, which may reflect stronger prior exposure to similar biomedical plots during pretraining. To test whether this gap on Sleep-QA reflects a limitation of the segment selection mechanism or of the input modality itself, we additionally instantiate ARTIST with a VLM backbone that consumes time series as plots. Under this representation, ARTIST surpasses TimeMaster on Sleep-QA, indicating that the gap is primarily modality-driven rather than caused by the controller-reasoner design (Appendix G).

### 4.2. Analysis of Time Series Data Utilization

Figure 3 examines how much of the time series ARTIST uses under RL and how this relates to accuracy on three datasets. Across datasets, greater coverage does not consistently improve performance. Instead, accuracy is highest when the model concentrates on a limited set of question-relevant segments, and it drops for questions that trigger near-complete sequence usage. For Sleep-QA and TRQA, the best accuracy occurs when the model uses roughly 30–50% of the signal, whereas questions that induce 90–100% utilization are substantially less accurate. TSQA exhibits a higher optimal range (about 70%), which is consistent with its shorter sequences and the need for proportionally broader coverage to capture relevant intervals.

### 4.3. Ablation Study

We conduct an ablation study to evaluate the core design choices of ARTIST, with results summarized in Table 3.

**Efficacy of Adaptive Segment Selection.** Removing the controller entirely, which processes the full time series as a static input, leads to a significant performance drop of 9.30% in average accuracy. This confirms that processing the entire sequence can be suboptimal, as irrelevant temporal noise may obscure salient evidence. In contrast, ARTIST's adaptive selection directs attention to task-relevant regions, enabling more precise reasoning.

**Joint Controller-Reasoner Optimization.** *Controller-only RL* configuration, where the reasoner remains frozen, results in an 8.94% decrease in average accuracy compared to the full model. This degradation arises from a distribution shift: during RL, the reasoner is conditioned on a dynamically selected set of segments chosen by the controller, whereas in SFT it was trained under a different input regime, receiving the full time series followed by a small number of additional segments. Freezing the reasoner therefore prevents it from adapting to the controller's evolving selection policy. Jointly updating both roles mitigates this mismatch and maintains alignment between segment selection and downstream reasoning.

**Reliability Reward.** The most substantial performance decline (21.44% average drop) occurs when removing the *Reliability Reward*. Relying on single-rollout accuracy is insufficient due to the inherent stochasticity of LLMs: a reasoner may produce a correct answer by chance even when the selected segments are incomplete or irrelevant. In these cases, a single successful rollout can mislead the controller during learning. Our reliability metric uses nested rollouts to measure answer consistency across repeated samples. This estimate reflects whether the selected segment set reliably supports correct reasoning, rather than producing an isolated correct outcome.

**Hierarchical Policy Optimization and Variance-Guided Sampling.** Replacing the trajectory-level objective with a myopic objective leads to a 12.28% performance drop. Here, the myopic objective refers to optimizing the controller based solely on the last interaction step, without propagating credit across the full interaction trajectory. With such an objective, the controller is encouraged to learn a greedy policy that is not optimized to make intermediate segment selections that lead to the best overarch-

*Table 1.* **Accuracy (%) and F1-score (%) comparisons with general LLMs and TS-based encoder baselines.**

| Model | ETI | | RCW | | ECG QA | | SLEEP QA | | TSQA | | TRQA | | Avg. | |
|---|---|---|---|---|---|---|---|---|---|---|---|---|---|---|
| | Acc | F1 | Acc | F1 | Acc | F1 | Acc | F1 | Acc | F1 | Acc | F1 | Acc | F1 |
| Random Guess | 25.00 | 25.00 | 50.00 | 50.00 | 50.00 | 50.00 | 16.67 | 16.67 | 29.67 | 29.67 | 37.13 | 37.13 | 34.74 | 34.74 |
| GPT-5 | | | | | | | | | | | | | | |
|    w/o statistics | 29.50 | 27.90 | 34.07 | 34.88 | 51.98 | 51.78 | 0.49 | 1.77 | 30.92 | 27.22 | 21.50 | 21.67 | 28.08 | 27.54 |
|    w/ statistics | 63.54 | 63.72 | 32.74 | 31.08 | 53.96 | 49.62 | 0.49 | 1.49 | 35.75 | 32.24 | 25.00 | 28.70 | 35.25 | 34.48 |
| Llama-3 8B | | | | | | | | | | | | | | |
|    w/o statistics | 25.50 | 25.96 | 43.97 | 40.41 | 50.00 | 48.45 | 31.99 | 14.09 | 48.97 | 46.75 | 57.69 | 53.88 | 43.02 | 38.26 |
|    w/ statistics | 35.50 | 34.39 | 62.83 | 45.82 | 50.00 | 48.90 | 22.06 | 13.34 | 44.93 | 44.12 | 55.50 | 53.02 | 45.14 | 39.93 |
| Qwen3-8B | | | | | | | | | | | | | | |
|    w/o statistics | 25.25 | 24.61 | 0.00[1] | 0.00[1] | 50.62 | 48.46 | 4.90 | 6.71 | 11.35 | 11.02 | 14.63 | 15.2 | 17.88 | 17.87 |
|    w/ statistics | 45.00 | 40.88 | 0.00[1] | 0.00[1] | 51.86 | 49.06 | 1.96 | 3.87 | 13.53 | 12.78 | 15.25 | 14.02 | 21.35 | 20.31 |
| Qwen3-14B | | | | | | | | | | | | | | |
|    w/o statistics | 32.00 | 30.66 | 33.19 | 27.40 | 50.99 | 50.42 | 3.43 | 5.73 | 24.15 | 22.50 | 33.00 | 29.54 | 29.46 | 27.08 |
|    w/ statistics | 42.00 | 47.34 | 22.57 | 22.91 | 54.58 | 51.79 | 3.43 | 4.48 | 24.15 | 22.20 | 29.00 | 30.32 | 29.29 | 29.84 |
| ChatTS-14B | | | | | | | | | | | | | | |
|    Base Model | 31.00 | 21.87 | 69.91 | 30.24 | 48.02 | 32.07 | 17.65 | 13.19 | 43.48 | 30.72 | 57.50 | 42.12 | 44.59 | 28.37 |
|    + SFT | 50.50 | 40.69 | 73.89 | 33.10 | 53.47 | 24.31 | 26.47 | 14.60 | 46.38 | 42.65 | 69.00 | 55.57 | 53.28 | 35.15 |
| OpenTSLM-4B | | | | | | | | | | | | | | |
|    + SFT | 82.69 | 82.66 | 65.49 | 38.29 | 69.50 | 41.00 | 35.37 | 18.99 | 47.50 | 35.81 | 76.25 | 69.36 | 62.80 | 47.68 |
| ITFormer-4B | | | | | | | | | | | | | | |
|    + SFT | 84.62 | 84.60 | 67.31 | 57.95 | 57.31 | 49.91 | 33.62 | 15.77 | 49.50 | 23.62 | 80.12 | 74.22 | 62.08 | 51.01 |
| ARTIST | | | | | | | | | | | | | | |
|    + SFT | 85.12 | 85.11 | 69.75 | **61.46** | 56.31 | **55.68** | 28.13 | 17.94 | 60.06 | 57.13 | 82.26 | 62.32 | 63.61 | 56.61 |
|    + SFT + RL | **87.03** | **87.10** | **77.00** | 50.00 | **69.81** | 52.67 | **36.63** | **19.21** | **62.00** | **58.66** | **83.06** | **78.02** | **69.26** | **57.61** |
|    **Improvement**[2] | **+2.41** | **+2.50** | **+3.11** | **+3.51** | **+3.14** | **+3.89** | **+1.26** | **+0.22** | **+12.50** | **+11.91** | **+2.94** | **+3.80** | **+6.46** | **+6.60** |

[1] Model failed to produce the required answer template and often repeated the input prompt; scores are reported as 0.00.
[2] Improvement denotes the performance gain over the strongest baseline for each dataset and metric.

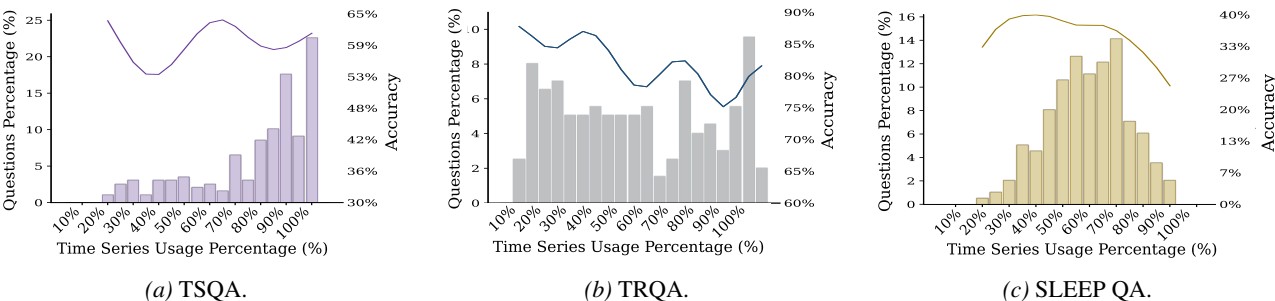

*(a)* TSQA.        *(b)* TRQA.        *(c)* SLEEP QA.

*Figure 3.* **Performance and question distribution across time-series usage levels.** Bars (left axis) show the percentage of questions falling into each usage-percentage bin, and the line (right axis) shows accuracy for questions in that bin.

ing segment set after multiple interactions. In contrast, our hierarchical objective trains the controller to optimize a sequence of segment selections that, together, forms an informative segment set. Replacing *Variance-guided Sampling* with standard stochastic sampling reduces average performance by 2.97%.

## 4.4. Illustration of Time Series Reasoning in ARTIST

Figure 4 illustrates how ARTIST addresses queries by selecting a targeted set of informative segments. In the apple sales case (Figure 4,left), the controller first retrieves an early segment to establish a baseline level. Guided by the

*Table 2.* **Accuracy (%) comparisons with vision models.**

| Model | ETI | RCW | ECG QA | SLEEP QA | TSQA | TRQA |
|---|---|---|---|---|---|---|
| Base Model | 27.00 | 50.33 | 53.54 | 49.44 | 52.51 | 66.09 |
| VL-Time | 28.50 | 57.52 | 58.91 | 22.06 | 44.93 | 53.50 |
|   + Few-shot | 22.00 | 48.67 | 64.36 | 23.53 | 38.16 | 58.50 |
| TimeMaster | 26.00 | 67.70 | 52.59 | 47.51 | 58.30 | 68.45 |
|   + RL | 49.00 | 76.99 | 69.31 | **72.55** | 61.22 | 72.08 |
| ARTIST | | | | | | |
|   + SFT | 85.12 | 69.75 | 56.31 | 28.13 | 60.06 | 82.26 |
|   + RL + SFT | **87.03** | **77.00** | **69.81** | 36.63 | **62.00** | **83.34** |

initial hypothesis of a potential trend shift, it then queries a late-sequence segment to test whether the change per-

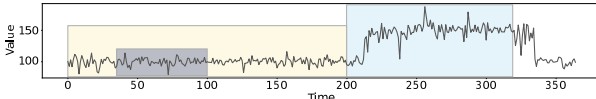

**Question:** What is the best description of this time series `<TS>`?
A) Daily hospital admissions for respiratory illnesses in a metropolitan city over one year, with a flu epidemic causing a noticeable increase in admissions.
B) Daily apple sales at a farmer's market stall over one year, with higher sales during the harvest season (August–November).
C) Daily counts of new employees joining a technology start-up over three years, with a mid-period recruitment drive leading to a sharp increase.
D) Minute-by-minute heart rate data of an individual, showing a significant spike between the 60th and 80th minute due to a sudden shock while watching a horror movie.

**Segment List:** [ s1: (0,199) , s2: (40,100) , s3: (200,320) ]

**Final reasoner trace:**
First, to solve the question, I need to establish a baseline to compare against later periods and assess whether a sustained change occurs. For this purpose, I'm choosing segment [0,199] because it is required to provide a reference period before any potential mid-to-late-year shift. From segment [0,199], I conclude that the series is steady with moderate variability and values mostly between about 85 and 110, indicating a stable baseline with no prolonged elevation. Next, I need to determine whether the latter part of the year shows a sustained change relative to the baseline. For this purpose, I'm choosing segment [200,300] because it is required to examine the end-of-year behavior to test for a persistent increase. From segment [200,300], I conclude that the series shifts to a higher level with greater variability, staying near 130-180 for an extended span and reaching the global maximum (~188), revealing a clear, sustained increase in the final months. The time series has 365 daily observations and exhibits a pronounced late-year uplift relative to the earlier baseline, consistent with a yearly external factor boosting the measured quantity in the final months.
**Answer: B**

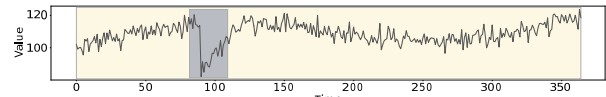

**Question:** What is the best description of this time series `<TS>`?
A) Weekly potato harvest levels from May–October on a small Idaho farm, tracked over 50 years (26 samples per year). External events such as unusually dry summers can sharply reduce yields.
B) Hourly counts of subjects tutored by an online tutoring service over one week, influenced by events such as school exams or holidays.
C) Daily bird counts in an urban park over one year to monitor population health, with a conservation event in April affecting observations.
D) Daily flu case counts at a busy city clinic over six months, with a new drug released at month three to assess changes before and after treatment.

**Segment List:** [ s1: (0,364) , s2: (80,115) ]

**Final reasoner trace:**
First, to solve the question, I need to determine the sampling horizon and whether a single contiguous change occurs within the year. For this purpose, I'm choosing segment [0, 364] because it is required to evaluate the entire sequence length and to check for a continuous month-long deviation across the full span. From this segment, I conclude that the series contains 365 points (consistent with daily sampling over one year) and exhibits one sustained shift roughly around indices 85–120, after which the series returns close to its earlier level. Next, I need to verify that the dip is truly a single contiguous event and not part of a broader shift. For this purpose, I select segment [80,115] centered on the suspected interval. From segment [80,115], I confirm a sustained reduction over roughly one month (about 40 consecutive days), followed by recovery. The evidence shows a daily series spanning a full year with a single month-long contiguous reduction, indicating a temporary external influence during part of the year.
**Answer: C**

*Figure 4.* **Examples of using ARTIST for reasoning with time series.** Source: Etiological time series reasoning bechmark.

*Table 3.* **Ablation on ECG-QA and RCW accuracy (%).**

| Method | ECG | RCW | Avg. |
|---|---|---|---|
| ARTIST | **69.81** | **77.00** | **73.41** |
| Reasoner Only | 65.33 | 62.88 | 64.11 |
| Controller-only RL | 60.81 | 68.13 | 64.47 |
| w/o Reliability Reward | 52.50 | 51.44 | 51.97 |
| w/o Trajectory-based Objective | 55.19 | 67.06 | 61.13 |
| w/o Variance-guided Sampling | 68.13 | 72.75 | 70.44 |

sists. The retrieved segment reveals a sustained level increase with higher variability, allowing the model to confirm a seasonal harvest uplift rather than a transient spike. In the bird count example (Figure 4,right), the model begins with a full-span segment to establish the sampling horizon and flag suspicious intervals. It then requests a higher-resolution segment around the anomaly to determine its duration and whether the signal recovers. This second segment shows a contiguous dip on the scale of months followed by a return to the prior level, which supports a temporary external influence rather than a persistent regime change. Segment selection yields a verifiable evidence trail that ties the final answer to specific temporal segments.

## 5. Conclusion

ARTIST is an RL approach that trains models to select task-relevant temporal segments while they reason toward an answer. Unlike prior methods that encode the full time series into a fixed representation, ARTIST interleaves segment selection with multi-step reasoning at inference time. On six benchmarks, ARTIST improves over strong LLMs, vision-language models, and time-series reasoning models.

**Limitations.** ARTIST incurs additional inference cost due to its iterative controller-reasoner interaction, which requires multiple model calls per question rather than a single forward pass. We quantify this cost relative to single-pass baselines in Appendix I. Finally, our evaluation focuses on univariate time-series tasks, extending the framework to multivariate signals and irregular sampling remains an important direction for future work.

## Impact Statement

This paper presents work whose goal is to advance the field of machine learning. There are many potential societal consequences of our work, none of which we feel must be specifically highlighted here.

## Acknowledgments

We gratefully acknowledge the support by NSF CAREER Award 2339524, ARPA-H Biomedical Data Fabric (BDF) Toolbox Program, Amazon Faculty Research, Google Research Scholar Program, AstraZeneca Research, GlaxoSmithKline Award, Roche Alliance with Distinguished Scientists (ROADS) Program, Sanofi iDEA-iTECH Award, Boehringer Ingelheim Award, Merck Award, Optum AI Research Collaboration Award, Pfizer Research, Gates Foundation (INV-079038), Chan Zuckerberg Initiative, John and Virginia Kaneb Fellowship at Harvard Medical School, Biswas Computational Biology Initiative in partnership with the Milken Institute, Harvard Medical School Dean's Innovation Fund for the Use of Artificial Intelligence, and the Kempner Institute for the Study of Natural

and Artificial Intelligence at Harvard University.

DISTRIBUTION STATEMENT A. Approved for public release. Distribution is unlimited. This material is based upon work supported by the Under Secretary of War for Research and Engineering under Air Force Contract No. FA8702-15-D-0001 or FA8702-25-D-B002.

Any opinions, findings, conclusions or recommendations expressed in this material are those of the author(s) and do not necessarily reflect the views of the Under Secretary of War for Research and Engineering.

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

# A. Controller and Reasoner Objectives and Rewards

## A.1. Controller Advantages

We form group-relative advantages (Shao et al., 2024) $\{\hat{A}_{\text{ctl}}^{(g)}\}_{g=1}^{G}$ as:

$$\mu_{\text{ctl}} = \frac{1}{G} \sum_{g=1}^{G} r_{\text{ctl}}^{(g)}, \qquad \sigma_{\text{ctl}} = \sqrt{\frac{1}{G} \sum_{g=1}^{G} (r_{\text{ctl}}^{(g)} - \mu_{\text{ctl}})^2}, \tag{14}$$

$$\hat{A}_{\text{ctl}}^{(g)} = \begin{cases} 0, & \sigma_{\text{ctl}} < \epsilon, \\ \dfrac{r_{\text{ctl}}^{(g)} - \mu_{\text{ctl}}}{\sigma_{\text{ctl}} + \epsilon}, & \text{otherwise,} \end{cases} \tag{15}$$

where $\epsilon$ is a small constant for numerical stability.

## A.2. Controller Objective

Let $\{o_{\text{ctl},i,t}^{(g)}\}_{t=1}^{T_i^{(g)}}$ be the sequence of the $T_i^{(g)}$ total tokens in the Controller output $(u_i^{(g)}, d_i^{(g)}, s_i^{(g)})$ at round $i$ of interaction trajectory $\tau^{(g)}$. For a batch of $B$ questions, the Controller objective function is defined as

$$\mathcal{J}_{\text{ctl}}(\theta) = \frac{1}{B} \sum_{b=1}^{B} \frac{1}{G} \sum_{g=1}^{G} \frac{1}{L^{(g)}} \sum_{i=1}^{L^{(g)}} \frac{1}{T_i^{(g)}} \sum_{t=1}^{T_i^{(g)}} \hat{A}_{\text{ctl}}^{(g)} \log \pi_\theta \left( o_{\text{ctl},i,t}^{(g)} \mid q_b, \{o_{\text{ctl},i,j}^{(g)}\}_{j=1}^{t-1} \right) \tag{16}$$

## A.3. Reasoner Advantages

We form group-relative advantages $\{\hat{A}_{\text{rsn}}^{(g^*,n)}\}_{n=1}^{N}$ as:

$$\mu_{\text{rsn}} = \frac{1}{N} \sum_{n=1}^{N} r_{\text{rsn}}^{(g^*,n)}, \qquad \sigma_{\text{rsn}} = \sqrt{\frac{1}{N} \sum_{n=1}^{N} (r_{\text{rsn}}^{(g^*,n)} - \mu_{\text{rsn}})^2}, \tag{17}$$

$$\hat{A}_{\text{rsn}}^{(g^*,n)} = \begin{cases} 0, & \sigma_{\text{rsn}} < \epsilon, \\ \dfrac{r_{\text{rsn}}^{(g^*,n)} - \mu_{\text{rsn}}}{\sigma_{\text{rsn}} + \epsilon}, & \text{otherwise.} \end{cases} \tag{18}$$

## A.4. Reasoner Objective

Let $\{o_{\text{rsn},t}^{(n)}\}_{t=1}^{T^{(n)}}$ be the sequence of the $T^{(n)}$ total tokens in the Reasoner output $(a_{L^{(g^*)}}^{(g^*,n)}, \hat{y}_{L^{(g^*)}}^{(g^*,n)})$ at round $L^{(g^*)}$ of Reasoner trajectory $\tau_{\text{rsn}}^{(g^*,n)}$. For a batch of $B$ questions, the Reasoner objective function is defined as

$$\mathcal{J}_{\text{rsn}}(\theta) = \frac{1}{B} \sum_{b=1}^{B} \frac{1}{N} \sum_{n=1}^{N} \frac{1}{T^{(n)}} \sum_{t=1}^{T^{(n)}} \hat{A}_{\text{rsn}}^{(n)} \log \pi_\theta \left( o_{\text{rsn},t}^{(n)} \mid q_b, \{o_{\text{rsn},j}^{(n)}\}_{j=1}^{t-1} \right) - D_{KL}(\pi_\theta || \pi_{\text{ref}}) \tag{19}$$

where the reference policy $\pi_{\text{ref}}$ is the reasoning model policy after SFT but before RL post-training.

## A.5. Combined Objective

The two objectives above are aggregated into a joint objective, which is maximized via gradient ascent:

$$\mathcal{J} = \mathcal{J}_{\text{ctl}} + \mathcal{J}_{\text{rsn}}. \tag{20}$$

## A.6. Format Reward

We introduce a format reward to ensure that both the Controller and the Reasoner produce well-structured outputs that follow the required interaction protocol. The format reward penalizes malformed, ambiguous, or incomplete responses and assigns a hard failure to critical violations.

**Reasoner Format Reward.** Given a reasoner completion $c$, we compute a scalar format score between -1 and 1. The score starts from 1 and is reduced according to the following cases:

- **Reasoning block:** The completion must contain exactly one `<think>` block. Missing or multiple reasoning blocks incur a penalty.

- **Answer block:** The completion must contain exactly one non-empty `<answer>` block. Missing answers constitute a critical violation.

- **Separation:** The `<answer>` block must not appear inside the `<think>` block; violations are penalized.

If a critical violation occurs (for example, missing answer), the format score becomes negative, triggering a hard failure.

**Controller Format Reward.** Let the controller produce a sequence of responses where the final response must terminate the interaction. Each controller iteration receives a per-step format score.

The trajectory level controller format score is computed by averaging the per iteration scores, unless a critical violation occurs. Specifically, the controller must satisfy the following rules:

- **Decision exclusivity:** Each step must contain exactly one decision: either a segment selection or an acceptance.

- **Iteration structure:** All non-final steps must issue a valid segment selection call, while the final step must accepts the reasoner's answer as final.

- **segment selection validity:** segment selections must be valid JSON and specify a time-series segment with valid bounds.

If any step violates these constraints, a violation indicator is triggered and the controller format score is set to $-1$.

**Hard Format Failure.** For both roles, any negative format score results in a hard penalty (-1).

## B. Datasets

*Table 4.* Overview of datasets.

| Dataset | Task | # Train | # Val | # Test | TS Len. ($\mu \pm \sigma$) | Domain |
|---------|------|---------|-------|--------|---------------------------|--------|
| RCW | Binary Choice | 19,135 | 4,405 | 226 | $4000 \pm 0$ | Nature |
| ECG QA | Binary Choice | 16,663 | 1,999 | 202 | $1000 \pm 0$ | ECG |
| SLEEP QA | Multi Choice | 7,268 | 1,817 | 204 | $1500 \pm 0$ | EEG |
| TSQA | Multi Choice | 7,243 | 1,811 | 207 | $22 \pm 5$ | Web/Nature/Healthcare/Energy |
| TRQA | Mixed (T/F, Multi Choice) | 17,241 | 2,487 | 200 | $136 \pm 65$ | Web/Transport/Finance/Energy/Sales/Nature |
| ETI | Multi Choice | 11,778 | 1,000 | 200 | $407 \pm 353$ | Sales/Energy/Entertainment/Tech/Transport/Health/Nature/Education |

We follow each benchmark's standard evaluation protocol, with minor restrictions for consistency across datasets.

**TSQA** (Kong et al., 2025a) is a large-scale time-series question answering dataset composed of real and synthetic data, designed to evaluate models' ability to understand and reason about temporal patterns beyond numeric prediction. We use its multiple-choice questions defined over univariate time series.

**RCW** is a univariate acoustic time-series dataset within the TimerBed benchmark (Liu et al., 2025b) that evaluates simple deterministic reasoning by requiring models to infer the presence of a North Atlantic right whale call from temporal–spectral structure in the signal.

**ECG-QA** (Oh et al., 2023) is a clinical time-series question answering dataset designed to evaluate medical reasoning over ECG signals, consisting of 10-second 12-lead ECG recordings paired with clinical context. In this work, we only use binary (true/false) questions involving a single lead, requiring models to reason over univariate cardiac time series.

**Sleep-QA** (Langer et al., 2025) is a clinical time-series question answering dataset designed to evaluate reasoning over sleep-stage dynamics, consisting of 30-second single-channel electroencephalogram (EEG) recordings annotated with sleep stages. Following prior work(Pouliou et al., 2025; Langer et al., 2025), non-REM stages 3 and 4 are merged, yielding five classes: Wake, REM, Non-REM1, Non-REM2, and Non-REM3.

**TRQA** (Jing et al.) is a large-scale time-series reasoning benchmark; we use its characterization task, which evaluates reasoning about intrinsic properties of univariate time series through true/false and multiple-choice questions.

**Etiological Reasoning (ETI)** (Merrill et al., 2024) is a time-series reasoning dataset that evaluates causal interpretation by requiring models to identify the most plausible generative scenario for a given univariate time series via multiple-choice questions. The original dataset consists of 6,978 synthetic time series; to obtain sufficient data for both RL and SFT, we additionally use a subset of the single–time-series (1TS) dataset released by the authors and adapt it to the etiological reasoning format using their provided code.

## C. SFT CoT Data Creation

Supervised fine-tuning reasoning traces follow a structured, interleaved format that alternates between natural-language reasoning and explicit time-series segment selection. Each trace consists of multiple reasoning steps, where intermediate reasoning motivates the selection of informative temporal segments, followed by a final answer. To construct these traces, we prompted GPT-5 to answer time-series questions conditioned on rendered visualizations. The resulting responses were then transformed into structured reasoning traces with explicit segment-selection steps and post-processed to ensure correctness, structural consistency, and adherence to the template. A simplified example is shown in Figure 1. Figure 2 presents the prompt used to generate the raw responses from which the structured reasoning traces were derived.

*Example 1.* SFT reasoning trace template.

```
<think> reasoning ... </think>
<timeseries_selection_tool> [x1, y1] </timeseries_selection_tool>
<think> reasoning ... </think>
<timeseries_selection_tool> [x2, y2] </timeseries_selection_tool>
<think> reasoning ... </think>
<answer> A / B / C / D / E </answer>
```

*Example 2.* SFT CoT curation example.

```
You are a world-class expert in time-series reasoning and analysis.

You are given:
- A time series (indexed from 0 to {ts_len - 1})
- A question about the time series
- Statistics about the time series
- The correct answer.

Your task has two parts:

(1) Identify the specific contiguous segments of the time series that are essential
↪  for solving the question.
- You may choose at most 3 segments.
- Each segment must be at least 8 time steps long (inclusive).
- Segment boundaries must lie within [0, {ts_len - 1}].
- Only include segments that are strictly necessary. If the entire series is needed,
↪  use a single segment [0, {ts_len - 1}].

(2) Using only the identified segments and the metadata, provide a concise,
↪  step-by-step explanation that leads to the correct answer.
```

```
Critical explanation constraint:
- For each segment, you MUST separate:
(a) The reasoning objective for selecting the segment, and
(b) The analysis of what the segment reveals.
- The sentence explaining why a segment is chosen MUST NOT describe, summarize, or
↪   analyze the values, patterns, trends, or variability within that segment, but
↪   explain the rationale and how to approach the question correctly.
- All observations and analyses of the segment MUST appear only in the subsequent
↪   sentence.

Important constraints:
- Base all explanations strictly on the observable behavior and statistical
↪   properties of the time series.
- Do NOT use external context, prior knowledge of labels, or textual descriptions of
↪   answer options.
- Do NOT mention or allude to any answer option or class label until the final
↪   answer line.

Your response MUST follow exactly the structure below, with no additional sections
↪   or deviations:

Important segments:
Segment 1: [x1, y1]
Explanation:
First, to solve the question, I need to <state the reasoning objective>.
For this purpose, I'm choosing segment [x1, y1] because it is required to <state the
↪   logical purpose this segment serves, without describing what is observed>.
From this segment, I conclude that <state the key insight derived from analyzing
↪   this segment>.

(If needed)
Segment 2: [x2, y2]
Explanation:
Next, I need to <state the reasoning objective>.
For this purpose, I'm choosing segment [x2, y2] is chosen because it is required to
↪   <state the logical purpose this segment serves, without describing what is
↪   observed>.
From this segment, I conclude that <state the key insight derived from analyzing
↪   this segment>.

(If needed)
Segment 3: [x3, y3]
Explanation:
Finally, I need to <state the reasoning objective>.
For this purpose, I'm choosing segment [x3, y3] is chosen because it is required to
↪   <state the logical purpose this segment serves, without describing what is
↪   observed>.
From this segment, I conclude that <state the key insight derived from analyzing
↪   this segment>.

Final conclusion:
One or two sentences that combine the insights from the selected segments to justify
↪   the correct answer, expressed purely in terms of time-series behavior.

Answer: X

Answer formatting rules:
- The final line must be exactly: "Answer: X"
- X must be a single capital letter:
- A/B/C/D for multiple-choice questions
- A/B for true/false questions
- The final line must contain nothing except the letter.
```

```
Question:
{question}

{option_block}

Additional data you may use (metadata):
Summary statistics:
- Mean: {mean}
- Min: {min_ts}
- Max: {max_ts}
- Std: {std}

Time series length: {ts_len}

Correct answer: {correct_ans}
```

## D. Supervised Fine-Tuning Configuration

Prior to reinforcement learning, we train ARTIST through large-scale pretraining following the procedure described in (Xie et al., 2024), enabling joint modeling of time-series representations and natural language inputs. For each benchmark, we then independently perform supervised fine-tuning using LoRA adapters (rank 8, $\alpha = 16$) for up to 5 epochs, with early stopping based on validation performance.

## E. RL Training Configuration

We train ARTIST using fully on-policy reinforcement learning with full-parameter fine-tuning. Optimization is performed using AdamW with learning rate $1 \times 10^{-6}$. Each update processes a batch of 64 question and time-series pairs. For each pair, $G = 6$ interaction trajectories are sampled for the controller, and for each trajectory, $N = 6$ final-round reasoner rollouts are generated. During rollout generation, we use a controller temperature of $1.0$, a reasoner temperature of $0.7$, and top-$p$ sampling with $p = 0.95$. We apply KL regularization only to the reasoner objective with a fixed coefficient $\beta = 0.002$, using the SFT-initialized model as the reference policy $\pi_{\mathrm{ref}}$. Training runs for one epoch over the dataset, and experiments are conducted on four NVIDIA H100 GPUs.

### E.1. Training Cost: RL vs. SFT

We report the wall-clock training cost of ARTIST's two stages on three benchmarks. The SFT stage uses LoRA adapters on a single H100, while the RL stage performs full-parameter fine-tuning across four H100s.

*Table 5.* **Training cost of ARTIST's SFT and RL stages.** Wall-clock and GPU-hours are reported per benchmark.

| Dataset | Stage | Trainable Params | GPUs | Wall-clock (h) | GPU-hours | Batch Size |
|---------|-------|------------------|------|----------------|-----------|------------|
| TRQA | SFT | LoRA (16.5M) | 1×H100 | 1.50 | 1.50 | 4 |
| | RL | Full (4B) | 4×H100 | 23.00 | 92.00 | 64 |
| ETI | SFT | LoRA (16.5M) | 1×H100 | 0.67 | 0.67 | 4 |
| | RL | Full (4B) | 4×H100 | 26.00 | 104.00 | 64 |
| ECG-QA | SFT | LoRA (16.5M) | 1×H100 | 1.35 | 1.35 | 4 |
| | RL | Full (4B) | 4×H100 | 27.00 | 108.00 | 64 |

The SFT stage converges within 2-3 epochs (early-stopped on validation accuracy) using a per-device batch size of 4, while the RL stage runs for around 100 updates with a global batch size of 64 question-time-series pairs.

## F. Prompts

*Example 3.* SFT Prompt.

```
You are a time series expert. Analyze ONLY the given time series data and answer the
↪   question.
# Output Schema
<think> your reasoning and how you came to the answer.</think>
<answer>
[Direct answer ONLY - the first line must be exactly one letter from the set of
↪   available options.]
</answer>
# Rules (MANDATORY)
- No text outside <think> and <answer>.
- In <think>, explain your reasoning and reference the time series.
- In <answer>, the first line must be exactly one letter from the set of available
↪   options.
# Time Series
### Segment 0: Timesteps [0, {ts_len}]
<TS_DATA>
# Question
{context}
```

*Example 4.* Reasoner Prompt.

```
You are a time series expert. Analyze ONLY the given time series data and answer the
↪   question.

# Output Schema
<think>One-two sentences describing your reasoning and how you came to the
↪   answer.</think>
<answer>
[Direct answer ONLY - the first line must be exactly one letter.
</answer>

# Rules (MANDATORY)
- No text outside <think> and <answer>.
- In <think>, explain you reasoning, how you came to the answer, and reference
↪   segments by number (e.g., "Seg 2 rises ~0.3 at t=150-200").
- if evidence is insufficient, state the most likely answer and reflect uncertainty
↪   (concisely. i.e I'm not sure about the answer because...).
- In <answer>:
- first line is exactly A, B, C, or D.

# Time Series Segments
{segments_section}

# Question
{question}
```

*Example 5.* Controller Prompt.

```
You are a time series analysis expert. Your task is to decide which time series
↪   segments an LLM needs to accurately answer a question related to a time series.

## Instructions
1. You receive the FULL time series embedding, the question, the segments that were
↪   given to the LLM so far, and the LLM's answer.
2. Review the question and the LLM's answer, and analyze the full time series
↪   embedding.
3. Evaluate if the LLM's answer is accurate and if the available segments provide
↪   sufficient information to answer the question accurately - you can try answering
↪   the question yourself to understand what is missing.
4. Make ONE of two decisions:
- Retrieve an additional segment: If you think that the answer is not accurate
↪   and/or there is critical information missing, use the tool to get an additional
↪   segment of time series data for the LLM.
- Accept the LLM's answer as final: If you think the LLM's answer is accurate and
↪   the information provided is sufficient.
```

```
## Full Time Series Embedding
<TS_DATA>

## Current Status
- Time series length: X timesteps
- Question: {question}
- Segments provided to the LLM so far:
{CURR_SEG_LIST_TEXT}

LLM's PREVIOUS ANSWER:
{REASONER_ANS}

## Output Format
You MUST respond in ONE of the following two formats:

Option 1: Retrieve an additional segment
Make a tool call asking for a specific segment of the time series data:
<think> includes your concise reasoning for why you are retrieving this segment and
↪   what information is missing </think>
<tool_call>
{{"name": "timeseries_selection_tool", "arguments": {{"ts_seg": [start, end]}}}}
</tool_call>

Option 2: Accept the LLM's answer as final
You should answer with the following format:
<think> includes your concise reasoning for why you are accepting the answer
↪   </think>
<answer>ACCEPT</answer>

## Important Rules
1. You can only call 'timeseries_selection_tool' ONCE per turn
2. Specify segments as integers: [100, 200] means timesteps 100 to 200 inclusive
4. Do NOT request segments outside valid range [0, {ts_len - 1}]
5. If you output "ACCEPT", the process will immediately conclude with the LLM's
↪   current answer

Make your decision now:
```

## G. Modality Analysis on Sleep-QA

Sleep-QA is the most challenging dataset in our analyis for reasoning models that consume the signal as a tokenized sequence, while vision-language baselines that operate on plotted signals perform better on it (Tables 1 and 2). This pattern suggests that the gap may stem from the input modality used to encode the EEG signal rather than from the segment-selection and reasoning mechanism itself. To test this hypothesis, we replace the ARTIST backbone (Qwen3-4B) with a vision-language backbone (Qwen2.5-VL-3B) and represent the time series as rendered plots rather than as a tokenized sequence. All other components are kept unchanged. For this analysis we perform supervised fine-tuning only; full SFT+RL training under the vision-based representation is left to future work.

Table 6 reports accuracy and F1 on Sleep-QA. Under the vision-based representation, ARTIST substantially outperforms both its tokenized-input counterpart and TimeMaster (the strongest baseline) under the same SFT-only training regime. This confirms that the performance gap on Sleep-QA is primarily driven by the input modality used to encode EEG signals, rather than by the segment selection or reasoning components of ARTIST.

*Table 6.* **Modality analysis on Sleep-QA (%).**

| Method | Acc | F1 |
|---|---|---|
| ARTIST (VLM backbone, SFT) | **65.00** | **39.22** |
| ARTIST (tokenized, SFT) | 28.13 | 17.94 |
| TimeMaster | 47.51 | 32.98 |

## H. Comparison with Dynamic Visual Search

To assess whether dynamic visual search methods developed for vision-language reasoning transfer to time series reasoning, we compare ARTIST against PixelReasoner (Su et al., 2026), a curiosity-driven pixel-space reasoning method that iteratively selects informative image regions. We adapt PixelReasoner to our setting by replacing its Qwen-VL backbone with the same Qwen3-4B backbone used by ARTIST, keeping the rest of its training and inference pipeline unchanged. We evaluate both methods on the ECG-QA dataset.

*Table 7.* **Comparison with dynamic visual search on ECG-QA (%).**

| Method | Acc | F1 |
|---|---|---|
| PixelReasoner | 60.15 | 51.52 |
| ARTIST | **69.81** | **52.67** |

## I. Inference Cost Analysis

ARTIST's controller-reasoner interaction issues multiple model calls per question, which introduces additional inference cost compared to baselines that process the time series in a single forward pass. We measure per-example inference time on TRQA, ETI, and ECG-QA against the two strongest fine-tuned baselines from Table 1 (OpenTSLM-4B and ITFormer-4B). All measurements are taken on a single NVIDIA H100 GPU.

Following our evaluation protocol, each example is evaluated with 8 independent runs to compute average accuracy and F1; reported times are aggregate per-example wall-clock over these 8 runs. Table 8 reports per-example inference time alongside accuracy and F1.

*Table 8.* **Per-example inference cost and accuracy on TRQA, ETI, and ECG-QA.**

| Dataset | Method | Acc | F1 | Time/example, 8 runs (min) |
|---|---|---|---|---|
| TRQA | ARTIST | **83.06** | **78.02** | 1.68 |
| | OpenTSLM-4B | 76.25 | 69.36 | 1.26 |
| | ITFormer-4B | 80.12 | 74.22 | 1.29 |
| ETI | ARTIST | **87.03** | **87.10** | 2.08 |
| | OpenTSLM-4B | 82.69 | 82.66 | 1.05 |
| | ITFormer-4B | 84.62 | 84.60 | 1.27 |
| ECG-QA | ARTIST | **69.81** | **52.67** | 2.00 |
| | OpenTSLM-4B | 69.50 | 41.00 | 0.78 |
| | ITFormer-4B | 57.31 | 49.91 | 0.98 |

## J. Scalability to Long Sequences

ARTIST is designed so that the reasoner conditions only on the controller-selected segments and the controller localizes task-relevant regions rather than encoding the full signal at each step. As a result, the computational cost of an interaction trajectory is dominated by the number and length of selected segments, rather than the total sequence length $H$. We test whether this design preserves ARTIST's selective-utilization benefits as the input becomes substantially longer than what was seen during training.

**Experimental setup.** We construct extended-length variants of RCW, our longest dataset (original length 4,000). For each test sample, we extend the time series to lengths 8,000 and 12,000 by appending tiled copies of the original signal beyond timestep 4,000, with small additive Gaussian noise. The original $[0, 4000)$ region remains unmodified and contains all task-relevant information, the appended region is uninformative. ARTIST is not re-trained for these longer inputs, so this experiment evaluates whether the controller generalizes its segment selection behavior to sequences three times longer than those seen during training. All results are averaged over 8 independent runs per sample.

**Results.** Table 9 shows that accuracy varies by less than 1.5 percentage points across sequence lengths from 4K to 12K, indicating that the controller continues to localize the task-relevant region $[0, 4000)$ even when surrounded by three times

as much uninformative signal. Per-example inference time increases by only 1.6% when tripling the sequence length, since ARTIST's computational cost is dominated by the controller-reasoner interaction over selected segments rather than by encoding the full time series.

*Table 9.* **ARTIST's performance and inference cost on RCW under extended sequence lengths.**

| TS Length | Acc | F1 | Time/example, 8 runs (min) |
|---|---|---|---|
| 4,000 | 77.00 | 50.00 | 1.880 |
| 8,000 | 75.66 | 51.21 | 1.895 |
| 12,000 | 76.11 | 48.37 | 1.910 |

This stability reflects a structural property of ARTIST's hierarchical design. The reasoner is optimized over a single final-round interaction and is conditioned only on controller-selected segments, so increasing $H$ does not change the reasoner's input distribution. The controller is optimized over multiple interaction rounds to select segments until the reasoner has enough information to answer confidently. Since the number of task-relevant regions in a time series is typically determined by the question rather than by $H$, extending the time series does not increase the expected number of segments the controller selects, and therefore does not substantially change the number of interaction rounds per question.

## K. Comparison with Time-Series Explainability Methods

Time-series explainability (XAI) methods such as TimeX (Queen et al., 2023) and TimeX++ (Liu et al., 2024b) identify timesteps relevant to a downstream prediction by producing continuous attribution maps over the input. While ARTIST produces discrete segment selections in service of question answering rather than post-hoc explanations of a separate predictor, the regions ARTIST's controller selects can be compared to the regions XAI methods identify as important. We perform this comparison on two synthetic benchmarks from the TimeX suite with ground-truth masks: FreqShape and SeqCombSingle.

**Evaluation protocol.** TimeX and TimeX++ produce continuous attribution maps, while ARTIST produces discrete segment selections that we represent as a binary mask (1 if a timestep lies inside any selected segment, 0 otherwise). Threshold-sweep metrics that are used in these studies are less suited to a direct comparison with ARTIST. We therefore evaluate on the following metrics, computed against the ground-truth masks:

- **Segment Hit Rate (SHR):** fraction of ground-truth regions that overlap with at least one selected segment.

- **GT Coverage (GTC):** fraction of ground-truth timesteps contained in selected segments.

- **Redundancy:** fraction of selected segments that do not overlap any ground-truth region.

- **Segment-level Precision/Recall/F1/IoU:** computed after expanding each ground-truth point by $\pm k$ timesteps ($k \in \{5, 10\}$), evaluating whether each method localizes the correct temporal neighborhoods rather than requiring exact pointwise matches.

- **Sufficient Evidence Rate (SER):** a task-specific metric that checks whether the selected regions contain enough ground-truth structure to answer the question. For FreqShape, this requires covering at least two consecutive spike clusters to determine the inter-spike interval. For SeqCombSingle, all discriminative subsequences must be covered, since the class label depends on their combination.

For TimeX and TimeX++, we binarize their soft masks at a threshold of $0.5$ before computing these metrics. We train both baselines on the original data splits (5-fold cross-validation) and evaluate all methods on the same 200-sample test subset. Results are reported as mean $\pm$ standard deviation across folds.

**Results.** Tables 10 and 11 report results on FreqShape and SeqCombSingle. ARTIST is competitive with both XAI baselines. On FreqShape, ARTIST achieves the best SHR, GTC, and redundancy, while remaining within 0.06 of TimeX++ on SER; in the segment-level metrics, TimeX and TimeX++ achieve higher precision but ARTIST's recall, F1, and IoU are substantially higher across both tolerance levels. On SeqCombSingle, ARTIST has lower SHR and SER than the XAI baselines but achieves the best GTC and redundancy, and dominates on segment-level precision, recall, F1, and IoU.

Overall, the comparison suggests that the segments ARTIST's controller selects to support question answering align well with regions identified as important by dedicated XAI methods, and exhibit complementary strengths: ARTIST tends to produce fewer, more concentrated selections that cover ground-truth regions tightly, while XAI methods produce broader attributions with higher precision but lower recall.

*Table 10.* **Region-level comparison on FreqShape.** Mean ± std across 5 folds. Bold marks the best value per column.

| Method | SHR ↑ | GTC ↑ | Redundancy ↓ | SER ↑ |
|---|---|---|---|---|
| ARTIST | **0.7340 ± 0.0205** | **0.6804 ± 0.0189** | **0.0138 ± 0.0062** | 0.6250 ± 0.0342 |
| TimeX | 0.6252 ± 0.0065 | 0.5600 ± 0.0066 | 0.0826 ± 0.0198 | 0.6100 ± 0.0141 |
| TimeX++ | 0.6430 ± 0.0092 | 0.5933 ± 0.0057 | 0.0601 ± 0.0217 | **0.6880 ± 0.0259** |

| Method | Tolerance ±5 | | | | Tolerance ±10 | | | |
|---|---|---|---|---|---|---|---|---|
| | Prec. | Rec. | F1 | IoU | Prec. | Rec. | F1 | IoU |
| ARTIST | 0.8407 | **0.6195** | **0.6872** | **0.5661** | 0.9522 | **0.5588** | **0.6796** | **0.5568** |
| TimeX | 0.9559 | 0.1854 | 0.2950 | 0.1795 | 0.9973 | 0.1559 | 0.2587 | 0.1554 |
| TimeX++ | **0.9753** | 0.1718 | 0.2810 | 0.1689 | **0.9988** | 0.1405 | 0.2384 | 0.1405 |

*Table 11.* **Region-level comparison on SeqCombSingle.** Mean ± std across 5 folds. Bold marks the best value per column.

| Method | SHR ↑ | GTC ↑ | Redundancy ↓ | SER ↑ |
|---|---|---|---|---|
| ARTIST | 0.4900 ± 0.0137 | **0.3406 ± 0.0134** | **0.0067 ± 0.0066** | 0.6467 ± 0.0172 |
| TimeX | 0.7095 ± 0.0149 | 0.1793 ± 0.0225 | 0.0274 ± 0.0218 | 0.6690 ± 0.0297 |
| TimeX++ | **0.7340 ± 0.0034** | 0.2006 ± 0.0238 | 0.0205 ± 0.0149 | **0.7180 ± 0.0067** |

| Method | Tolerance ±5 | | | | Tolerance ±10 | | | |
|---|---|---|---|---|---|---|---|---|
| | Prec. | Rec. | F1 | IoU | Prec. | Rec. | F1 | IoU |
| ARTIST | **0.9310** | **0.2275** | **0.3537** | **0.2272** | **0.9367** | **0.1707** | **0.2794** | **0.1707** |
| TimeX | 0.7299 | 0.1087 | 0.1830 | 0.1066 | 0.7314 | 0.0809 | 0.1414 | 0.0796 |
| TimeX++ | 0.7372 | 0.1223 | 0.2025 | 0.1195 | 0.7398 | 0.0916 | 0.1576 | 0.0899 |

