# OpenReview forum: "Adaptive Time Series Reasoning via Segment Selection"
_ICML.cc/2026/Conference — ICML 2026 regular_

### Official Review · Reviewer_HBWX · 2026-03-06

**Soundness:** 3
**Presentation:** 3
**Significance:** 3
**Originality:** 3
**Overall Recommendation:** 4
**Confidence:** 3

**Summary:**

ARTIST is a time-series reasoning framework that interleaves natural-language reasoning with adaptive temporal segment selection by treating the data as an active resource. It utilizes a controller-reasoner architecture where a high-level policy identifies informative segments and a low-level policy generates answers conditioned on that specific evidence. This dual-role system is optimized through hierarchical reinforcement learning, employing rewards for correctness and reliability to refine the model's ability to locate task-relevant information. ARTIST achieve better performance than the state-of-the-art models.

**Compliance With Llm Reviewing Policy:**

Affirmed.

**Final Justification:**

Based on my previous comment regarding the comparison with XAI methods, the authors have conducted additional experiments, which I appreciate. However, one aspect I am not fully convinced by is their evaluation protocol. In computing the metric, the authors use all segments selected by ARTIST across its reasoning and segment selection iterations. I believe a more appropriate comparison would be to consider only the segment selected in the final iteration of ARTIST.

Overall, I maintain my score at 4.

**Key Questions For Authors:**

- What do the variables **a** and **u** represent?

- What is the rationale behind the summation used in Equation 2?

- What is the impact of the parameters **G** and **N**, introduced in Section 3.5, on the performance of ARTIST?

- It would strengthen the paper to compare the segments selected by ARTIST for decision-making with those highlighted by time-series XAI methods such as TimeX, TimeX++, and TimeSliver. The authors could use a few synthetic datasets (where the ground-truth important segments are known) to demonstrate ARTIST’s ability to identify important segments and to evaluate how well it aligns with existing XAI approaches.

**Limitations:**

Authors have discussed the limitations.

**Strengths And Weaknesses:**

Strengths:

•	The paper evaluates the method against strong baselines and across multiple datasets. The results are encouraging.

•	A detailed ablation study is provided to analyze the contribution of different components of the proposed method.

Weaknesses:

•	Several equations and variables are not clearly introduced or defined (see questions below).

•	The effects of some variables are not discussed in sufficient detail (see questions below).

---

> ### Author Rebuttal · Authors · 2026-03-31
>
> Thank you for your helpful feedback and support! We are grateful that you acknowledge our strong results and our experimental validation and ablation studies.
>
> We have provided responses to your comments below and have included results from new experiments that we hope address your concerns and make our study stronger and more insightful. If you find that your concerns are addressed after reviewing our responses, we would greatly appreciate it if you considered raising your score. If there are any remaining questions, please let us know and we would be happy to address them!
>
> ---
>
> ## W1 + Q1-2 : Clarifying notations
> We thank the reviewer for pointing out the lack of clarity in notation and equation definitions. This is a writing issue, and we have revised the manuscript to more clearly introduce and define all variables.
>
> In particular, $u$ denotes the controller’s reasoning trace at each interaction step (not including the selected segments), while $a$ denotes the reasoner’s reasoning trace conditioned on the selected segments (not including the final answer). We now explicitly define these variables when they are first introduced.
>
> Regarding the rationale behind the summation in Eq. (2), the equation is describing that the probability of producing some final response $Y$ given a query $(q, T)$ is the expectation (weighted sum by probability) of producing the response $Y$ given a reasoning trace $z$ and query $(q, T)$ across all possible reasoning traces $z \in \mathcal{Z}$.
>
> ## W2 + Q3: Impact of Controller and Reasoner Rollout Sizes (G and N)
> We thank the reviewer for this question. The parameters G (number of controller trajectories) and N (number of reasoner rollouts per trajectory) specify the quality of the reward signals and advantage estimates.
> We evaluate sensitivity to these parameters by varying their values on dataset TRQA, using the same setting for both G and N for simplicity and due to computational constraints. We note that increasing group size does not significantly impact performance.
>
> |G|N|Acc|F1|
> |---|---|---|---|
> |2|2|82.63|77.04|
> |6|6|83.06|78.02|
> |8|8|83.25|78.10|
>
> ## Q4: Comparison of ARTIST Segment Selection with XAI Approaches
> We thank the reviewer for this great suggestion. Comparing ARTIST with time-series XAI methods is not straightforward and would require substantial adaptations to ensure a fair evaluation. In particular, existing methods such as TimeX++ produce soft masks and are evaluated using metrics such as AUPRC/AUP/AUR, whereas ARTIST produces discrete segment selections. These differences in the output representation would require careful alignment of evaluation protocols.

---

> > ### Author Rebuttal · Reviewer_HBWX · 2026-04-01
> >
> > I want to thank the authors for their response. The comparison with XAI methods is still missing, so I will keep my score the same.

---

> > > ### Author Response · Authors · 2026-04-06
> > >
> > > We thank the reviewer for this valuable suggestion. Following the feedback, we have conducted the requested comparison.
> > >
> > > We evaluate ARTIST against TimeX (Queen et al. 2023) and TimeX++ (Liu et al. 2024) on two synthetic datasets with known ground-truth explanations: FreqShape (spike frequency/shape identification) and SeqCombSingle (subsequence combination identification), both from the TimeX benchmark.
> > >
> > > ### Evaluation protocol
> > > TimeX and TimeX++ produce attribution maps that assign continuous importance values to individual timesteps, while ARTIST produces discrete segment selections, i.e., contiguous intervals chosen during reasoning, that would translate to a binary map (1 if the timestep is part of a selected segment, 0 otherwise). Standard metrics used by TimeX (AUPRC, AUP, AUR) sweep across thresholds and thus are more meaningful for continuous maps like those produced by TimeX than the discrete segments selected in ARTIST. Therefore, we evaluate on the following metrics instead:
> > >
> > > (1) **Segment Hit Rate (SHR)**: The fraction of ground-truth (GT) regions that overlap with at least one selected segment.
> > >
> > > (2) **GT Coverage (GTC)**: The fraction of GT timesteps falling inside selected segments.
> > >
> > > (3) **Redundancy**: The fraction of selected segments that do not overlap any GT region.
> > >
> > > (4) **Segment-level GT metrics**: Precision, recall, F1, and IoU computed after expanding each GT point by ±k timesteps, evaluating whether methods localize the correct temporal neighborhoods rather than requiring exact pointwise matches.
> > >
> > > (5) **Sufficient Evidence Rate (SER)**: A task-specific metric that checks whether the selected segments contain enough GT regions to actually answer the question. For FreqShape, this requires covering at least two consecutive spike clusters, which is sufficient to determine the inter-spike interval. For SeqComb, all discriminative subsequences must be covered, since the class label depends on their combination.
> > >
> > > For TimeX and TimeX++, we binarize their soft masks at a threshold of 0.5 before computing these metrics. We trained both baselines on the original data splits (5-fold cross-validation) and evaluated all methods on the same 200 test samples (subset of original test dataset due to time constraint).
> > >
> > > ### Results
> > >
> > > Results are provided in the tables below, and will be included in the final version of the paper. Overall, ARTIST is very competitive with the xAI methods. For the FreqShape dataset, ARTIST achieves a superior SHR and GTC compared to TimeX/TimeX++, while also being much less redundant. In terms of SER, ARTIST and TimeX are comparable, while TimeX++ has an advantage. For segment-level metrics, while TimeX/TimeX++ achieve somewhat better precision, ARTIST achieves *drastically* better recall and F1, as well as drastically better IoU. For the SeqCombSingle dataset, ARTIST achieves superior GTC and redundancy, but lags in SHR and SER. However, ARTIST achieves superior performance across the board for segment-level metrics (precision, recall, F1, IoU) on this dataset.
> > >
> > > **FreqShape:**
> > >
> > > |Method|SHR|GTC|Redundancy|SER|
> > > |---|---|---|---|---|
> > > |ARTIST|**0.7340 ± 0.0205**|**0.6804 ± 0.0189**|**0.0138 ± 0.0062**|0.6250 ± 0.0342|
> > > |TimeX|0.6252 ± 0.0065|0.5600 ± 0.0066|0.0826 ± 0.0198|0.6100 ± 0.0141|
> > > |TimeX++|0.6430 ± 0.0092|0.5933 ± 0.0057|0.0601 ± 0.0217|**0.6880 ± 0.0259**|
> > >
> > > **Segment-Level GT - Tolerance ±5**
> > >
> > > |Method|Precision|Recall|F1|IoU|
> > > |---|---|---|---|---|
> > > |ARTIST|0.8407 ± 0.0152|**0.6195 ± 0.0177**|**0.6872 ± 0.0168**|**0.5661 ± 0.0170**|
> > > |TimeX|0.9559 ± 0.0129|0.1854 ± 0.0133|0.2950 ± 0.0133|0.1795 ± 0.0106|
> > > |TimeX++|**0.9753 ± 0.0111**|0.1718 ± 0.0075|0.2810 ± 0.0074|0.1689 ± 0.0060|
> > >
> > > **Segment-Level GT - Tolerance ±10**
> > >
> > >
> > > |Method|Precision|Recall|F1|IoU|
> > > |---|---|---|---|---|
> > > |ARTIST|0.9522 ± 0.0143|**0.5588 ± 0.0168**|**0.6796 ± 0.0169**|**0.5568 ± 0.0168**|
> > > |TimeX|0.9973 ± 0.0023|0.1559 ± 0.0137|0.2587 ± 0.0171|0.1554 ± 0.0133|
> > > |TimeX++|**0.9988 ± 0.0005**|0.1405 ± 0.0080|0.2384 ± 0.0104|0.1405 ± 0.0080|
> > >
> > >
> > > **SeqCombSingle:**
> > >
> > > |Method|SHR|GTC|Redundancy|SER|
> > > |---|---|---|---|---|
> > > |ARTIST|0.4900 ± 0.0137|**0.3406 ± 0.0134**|**0.0067 ± 0.0066**|0.6467 ± 0.0172|
> > > |TimeX|0.7095 ± 0.0149|0.1793 ± 0.0225|0.0274 ± 0.0218|0.6690 ± 0.0297|
> > > |TimeX++|**0.7340 ± 0.0034**|0.2006 ± 0.0238|0.0205 ± 0.0149|**0.7180 ± 0.0067**|
> > >
> > > **Segment-Level GT (Tolerance ±5)**
> > >
> > >
> > > |Method|Precision|Recall|F1|IoU|
> > > |---|---|---|---|---|
> > > |ARTIST|**0.9310 ± 0.0204**|**0.2275 ± 0.0103**|**0.3537 ± 0.0134**|**0.2272 ± 0.0103**|
> > > |TimeX|0.7299 ± 0.0153|0.1087 ± 0.0139|0.1830 ± 0.0198|0.1066 ± 0.0129|
> > > |TimeX++|0.7372 ± 0.0085|0.1223 ± 0.0142|0.2025 ± 0.0204|0.1195 ± 0.0140|
> > >
> > > **Segment-Level GT (Tolerance ±10)**
> > >
> > >
> > > |Method|Precision|Recall|F1|IoU|
> > > |---|---|---|---|---|
> > > |ARTIST|**0.9367 ± 0.0196**|**0.1707 ± 0.0084**|**0.2794 ± 0.0115**|**0.1707 ± 0.0084**|
> > > |TimeX|0.7314 ± 0.0144|0.0809 ± 0.0106|0.1414 ± 0.0162|0.0796 ± 0.0100|
> > > |TimeX++|0.7398 ± 0.0068|0.0916 ± 0.0109|0.1576 ± 0.0169|0.0899 ± 0.0108|

---

### Official Review · Reviewer_s6PF · 2026-03-11

**Soundness:** 3
**Presentation:** 3
**Significance:** 3
**Originality:** 3
**Overall Recommendation:** 4
**Confidence:** 4

**Summary:**

This paper proposes ARTIST, a framework for time series reasoning that formulates the problem as a sequential decision process. Instead of encoding the full time series into a fixed representation, ARTIST uses a controller-reasoner architecture where a high-level controller iteratively selects temporal segments and a low-level reasoner performs analysis conditioned on the accumulated segments. Both roles are instantiated from a single policy model via role-specific prompting. The model is trained in two stages: supervised fine-tuning on curated interaction traces, followed by collaborative self-play reinforcement learning. The RL stage uses a hierarchical optimization scheme with nested rollouts — the controller is optimized using a reliability-based reward (measuring the reasoner's consistency across multiple samples given the selected segments), while the reasoner is optimized using correctness reward with a variance-guided sampling strategy for efficiency. The authors evaluate on six time series reasoning benchmarks across clinical, financial, and environmental domains, reporting an average improvement of 6.46 percentage points over the strongest baseline while using only 30-70% of the input time series.

**Compliance With Llm Reviewing Policy:**

Affirmed.

**Final Justification:**

I am satisfied with the authors rebuttal so I raise the score to 4.

**Key Questions For Authors:**

1. **On the critical violation indicator.** The indicator $\mathbb{I}_{\text{viol}}$ plays a decisive role in the controller reward, triggering a hard penalty of $-1$ that overrides any reliability signal. Yet its definition is deferred entirely to the appendix. Could the authors provide a concise definition in the main text? What specific conditions constitute a critical violation versus a minor format error captured by $f_i$?

2. **On the univariate limitation.** The paper restricts the setting to univariate time series ($K=1$). Real-world applications in clinical monitoring, finance, and environmental sensing almost universally involve multivariate signals. Could the authors discuss the feasibility of extending ARTIST to the multivariate setting? Does the controller's segment selection mechanism generalize naturally when multiple co-evolving variables must be considered jointly?

3. **On variance-guided sampling efficiency.** The reasoner is updated using only a single group $g^*$ selected by variance-guided sampling. This is an aggressive approximation — with small $G$, the reasoner receives very limited training signal per update. Have the authors compared this against using top-$k$ groups or all $G$ groups? How sensitive is training stability to this design choice?

**Limitations:**

yes

**Strengths And Weaknesses:**

Strengths:

1. Well-motivated problem formulation. The shift from static, fixed-view time series encoding to dynamic, question-adaptive segment selection is intuitive and well-justified.

2. Technically coherent framework with several meaningful design choices. The controller-reasoner architecture provides a clean separation between "where to look" and "how to reason." The reliability reward for the controller (measuring reasoner consistency across multiple rollouts rather than single-sample correctness) is a thoughtful design that provides a more stable learning signal for evaluating segment selection quality.

3. Solid empirical validation. The evaluation spans six benchmarks across diverse domains (clinical, financial, environmental), and the reported improvements over multiple baselines (LLMs, VLMs, and prior time series reasoning systems) are substantial.

Weakness:
1. Insufficient discussion of related work on dynamic visual search in VLMs. The idea of iteratively selecting and attending to informative regions conditioned on intermediate reasoning is well-established in the vision-language model literature (e.g., visual search, iterative attention, region zoom-in mechanisms). The paper positions ARTIST primarily against static time series encoding methods and self-play RL, but does not discuss or differentiate from these closely related VLM approaches. Given that the core contribution is essentially adaptive information selection during inference, a thorough comparison with dynamic visual grounding and search methods is necessary to clarify the novelty of ARTIST in the broader context.

2. Imprecise framing in the introduction and misalignment between claims and method. The introduction characterizes PPO, GRPO, and DAPO as "token-level RL" methods, which is inaccurate — GRPO and DAPO operate with sequence-level rewards and do not perform per-token credit assignment. The stated motivation that ARTIST addresses the token-level credit assignment problem is not well-supported by the actual method, which essentially applies GRPO-style group-relative advantages at two levels (controller trajectories and reasoner rollouts). The true contribution appears to be role-level disentanglement rather than resolving a token-level optimization issue. This misalignment between the framing and the technical contribution weakens the narrative coherence of the paper.

3. Missing controlled baselines with equivalent training data. The experimental comparison would be significantly strengthened by fine-tuning baseline models (e.g., Qwen-series models) on the same supervised data used for ARTIST's SFT stage. Without this controlled comparison, it is unclear how much of ARTIST's improvement comes from the proposed architecture and RL training versus simply having access to curated time series reasoning traces.

---

> ### Author Rebuttal · Authors · 2026-03-31
>
> Thank you for your feedback! We are happy that you acknowledge the value of shifting from static encodings to adaptive segment selection, and that you found our role-disentanglement design thoughtful and meaningful. We also appreciate your acknowledgement of our strong empirical results and comprehensive validation.
>
> We provide detailed responses and new experimental results below to address your comments. If you find that your concerns are addressed, we would greatly appreciate it if you considered raising your score. If any questions remain after reviewing these responses, we would be happy to address them!
>
> ---
> ## W1: Relation to Dynamic Visual Search
> We agree that connections to dynamic visual search in VLMs should be discussed more explicitly, and we will revise the related work section to include these approaches.
>
> While ARTIST shares high-level similarity with VLMs that iteratively select informative regions, there are fundamental differences in both the problem setting and learning signal. Vision tasks typically involve identifying localized entities with clear spatial boundaries, often supported by explicit supervision signals (e.g., bounding boxes). In contrast, time-series (TS) reasoning does not provide explicit segment-level supervision; ARTIST must learn segment selection through weak supervision from reasoning outcomes rather than relying on predefined targets.
>
> We conducted two new experiments related to VLMs. First, we evaluated an ARTIST SFT variant using a VLM backbone (refer to our response W2 for Reviewer ifFu). Second, we trained a PixelReasoner model (replacing Qwen-VL with a Qwen3-4B backbone) and evaluated it on ECG-QA (results below). ARTIST achieves a stronger performance, indicating that transferring visual search mechanisms to TS reasoning is not sufficient.
>
> ||Acc|F1|
> |---|---|---|
> |PixelReasoner|60.15|51.52|
> |ARTIST|**69.81**|**52.67**|
>
> ## W2: Clarifying “Token-level” and ARTIST’s Hierarchical RL Strategy
> We apologize that the introduction was a bit imprecise in wording, and we will revise it in the final version. We would like to clarify our initial choice of wording. By “token-level RL” we did not mean that the advantages are calculated at a token-level granularity (we are aware GRPO/DAPO advantages are sequence level). Instead, we meant that the sequential decision making (SDM) problem for reasoning models trained using these approaches is formulated such that a single action/timestep corresponds to a single token, and all tokens are considered during the calculation of the parameter update. This is opposed to our hierarchical optimization, where in addition to the token-level SDM, we also optimize a higher-level SDM problem at the interaction level where a timestep corresponds to an interaction iteration and an action corresponds to a controller response that leads to a segment selection (with the higher-level update only considering tokens corresponding to controller outputs and not reasoner outputs, decoupling credit assignment to these segment selection actions from reasoning tokens).
>
> ## W3: Controlled Baselines with Equivalent Training Data (Clarification)
> We realize that this may not have been sufficiently clear. Please note that we already include baselines that are fine-tuned using the same supervised data as ARTIST’s SFT stage (the “+SFT” rows in Table 1). We will revise the manuscript to make this more explicit.
>
> ## Q1: Clarifying the Format Reward
> The definition of the critical violation indicator was defined in Appendix A.6, but will make it clearer in the main paper. The critical violation indicator is triggered when the output is non-executable, while minor format errors are executable but contain imperfect outputs, e.g., multiple reasoning (<think>) blocks.
>
> ## Q2: Extension to Multivariate Time Series
> There is no reason preventing ARTIST from being applied to multivariate TS. ARTIST can make segment selections to multivariate TS, but segment boundaries would be the same across all variables (variables considered jointly). However, if it is desirable to select different segments for each variable, extensions would be required. Exploring this is a valuable future direction, but our current work primarily serves to establish a new method for TS reasoning that focuses on adaptive selection/role disentanglement, so we limit the scope of this work to univariate TS.
>
> ## Q3: Variance-Guided Sampling (VGS)
> We thank the reviewer for the suggestion of selecting the top k>1 groups for the reasoner update. We tried this idea initially; however, we could only choose k=1 due to GPU memory limitations (using k groups would result in k x N rollouts).
>
> We chose VGS over uniform random sampling to maximize the learning signal. We evaluate the impact of VGS in our ablation study (Table 3, reproduced below). Removing VGS results in a performance drop, demonstrating its value.
>
> ||ECG|RCW|Avg.|
> |---|---|---|---|
> |Variance-guided|69.81|77|73.41|
> |Uniform|68.13|72.75|72.75|

---

> > ### Author Rebuttal · Reviewer_s6PF · 2026-04-04
> >
> > Thanks for the clear rebuttal, in which my concerns are fully addressed.

---

### Official Review · Reviewer_WcoD · 2026-03-13

**Soundness:** 2
**Presentation:** 3
**Significance:** 2
**Originality:** 3
**Overall Recommendation:** 4
**Confidence:** 4

**Summary:**

This paper studies time-series reasoning tasks where models must answer natural language questions based on time-series data. A key challenge is that relevant evidence may appear only in specific temporal segments, while most existing approaches compress the entire time series into a fixed representation before inference. To address this limitation, the paper proposes ARTIST, a framework that formulates time-series reasoning as a sequential decision process. ARTIST introduces a controller–reasoner architecture that adaptively selects informative temporal segments during inference. The controller chooses which segments of the time series to inspect, while the reasoner processes the selected information to answer the question. The segment selection policy is trained using reinforcement learning with answer correctness as the optimization signal. Experiments on six time-series reasoning benchmarks show that ARTIST improves performance over strong baselines while using only a fraction of the input time series.

**Compliance With Llm Reviewing Policy:**

Affirmed.

**Final Justification:**

The rebuttal and follow-up responses substantially improve the empirical evaluation and address several of my concerns. In particular, the additional ablation studies clarify the contribution of each component, and the detailed analysis of computational cost provides a clearer picture of the trade-offs.

Regarding the main concern on scalability, I appreciate the additional experiments on extended sequence lengths. While the evaluation is based on synthetically extended sequences, the results consistently demonstrate stable performance and minimal increase in inference cost, which provides reasonable empirical support for the claimed scalability behavior.

Overall, the additional evidence and clarifications significantly strengthen the paper. Accordingly, I revise my score from 3 to 4.

**Key Questions For Authors:**

-	Can the authors provide ablation studies that isolate the contribution of the controller, segment selection policy, and reasoning module? Such analysis would help clarify which components are primarily responsible for the observed performance improvements.

-	How does the proposed method scale to longer or higher-resolution time-series data? Additional experiments or analysis evaluating performance on substantially longer sequences would strengthen the empirical validation.

**Limitations:**

No. The paper does not explicitly discuss the limitations of the proposed approach. The authors could add a brief discussion on the scalability of the method and the potential computational trade-offs introduced by sequential segment selection.

**Strengths And Weaknesses:**

Strengths

-	The paper addresses an interesting and increasingly relevant problem: time-series reasoning with natural language queries.

-	The proposed segment selection formulation is intuitive and provides a natural mechanism for question-adaptive analysis of time-series data.

-	The controller–reasoner architecture is conceptually clear and can potentially be applied to other temporal reasoning tasks.

Weaknesses

-	The method combines multiple elements (controller, RL-based segment selection, and reasoning module), but the paper does not provide sufficient ablation studies to clearly identify which components contribute most to the performance gains.

-	While the method is motivated by the challenge of reasoning over long time-series data, the experiments do not sufficiently demonstrate how the approach scales as the sequence length or resolution increases.

-	The proposed sequential segment selection process may introduce additional inference cost, but the paper does not provide a detailed analysis of the computational trade-offs compared to baseline approaches.

---

> ### Author Rebuttal · Authors · 2026-03-31
>
> Thank you for your insights and feedback! We appreciate your acknowledgement of the high increasing relevance of time series reasoning, and we are glad you found our adaptive segment selection mechanism and controller-reasoner architecture to be intuitive, clearly motivated, and generalizable.
>
> Below, we have provided responses to your comments and have included results from additional experiments that we hope address your concerns and make our study stronger and more insightful. If you find that your concerns are addressed after reviewing our responses, we would greatly appreciate it if you considered raising your score. If there are any remaining questions, please let us know and we would be happy to address them!
>
> ---
> ## W1+Q1: Ablation Analysis of ARTIST Components
> We thank the reviewer for pointing out the need to better disentangle the contributions of the controller, RL-based segment selection, and reasoning module. We note that we already include ablation studies in Table 3, and we summarize them here for clarity in the table below. The results show that each component contributes significantly to performance.
> We will revise the manuscript to make these component-wise contributions more explicit.
> |Method Variant|Controller|RL-based Opt.|Reasoning Module|ECG-QA Acc (%)|RCW Acc (%)|Avg.|
> |---|---|---|---|---|---|---|
> |ARTIST (full)|V|V|V|**69.81**|**77.00**|**73.41**|
> |Reasoner only training during RL|X|V|V|65.33|62.88|64.11|
> |Frozen reasoner during RL (Controller training only)|V|V|V|60.81|68.13|64.47|
> |w/o RL (SFT-only)|V|X|V|56.31|69.75|63.03|
> |w/o Reliability Reward|V|V|V|52.50|51.44|51.97|
> |w/o Trajectory-level objective|V|V|V|55.19|67.06|61.13|
> |w/o Variance-guided sampling|V|V|V|68.13|72.75|70.44|
>
> ## W2+Q2: Evaluation Across Varying Sequence Lengths (Including Long Time Series)
> We thank the reviewer for raising this point. ARTIST is evaluated across multiple datasets spanning a wide range of sequence lengths, from short to long time series. In particular, the RCW dataset contains sequences of length 4000, which is commonly considered long in time-series reasoning benchmarks [1,2,3,4,5]. If the reviewer believes there is a benchmarking dataset for time series reasoning (involving complex tasks expressed in natural language) consisting of time series with length longer than 4000 that we should evaluate on, we would be happy to take suggestions and include the results in the camera-ready version of the paper.
>
> ## W3: Inference Cost and Computational Trade-offs
> We thank the reviewer for raising the computational trade-offs of the sequential segment selection process.
>
> ### Inference Cost
> TRQA
> |Method|Acc|F1|Time/example (min)|#GPUs|
> |---|---|---|---|---|
> |ARTIST|**83.06**|**78.02**|1.68|1|
> |OpenTSLM|76.25|69.36|1.26|1|
> |ITFORMER|80.12|74.22|1.29|1|
>
> ETI
> |Method|Acc|F1|Time/example (min)|#GPUs|
> |---|---|---|---|---|
> |ARTIST|**87.03**|**87.10**|2.08|1|
> |OpenTSLM|82.69|82.66|1.05|1|
> |ITFORMER|84.62|84.60|1.27|1|
>
> ECG-QA
> |Method|Acc|F1|Time/example (min)|#GPUs|
> |---|---|---|---|---|
> |ARTIST|**69.81**|**52.67**|2.00|1|
> |OpenTSLM|69.50|41.00|0.775|1|
> |ITFORMER|57.31|49.91|0.98|1|
>
>
>
> ### Training Cost and Convergence of RL vs. SFT
> TRQA
>
> |Stage|Trainable Params|GPUs|Wall-clock (h)|GPU-hours|Updates/Epochs|Batch Size|
> |---|---|---|---|---|---|---|
> |SFT|LoRA (16.5 M)|1×H100|1.5|1.5|3|4|
> |RL|Full (4B)|4×H100|23|92|100|64|
>
>
> ETI
>
> |Stage|Trainable Params|GPUs|Wall-clock (h)|GPU-hours|Updates/Epochs|Batch Size|
> |---|---|---|---|---|---|---|
> |SFT|LoRA (16.5 M)|1×H100|0.667|0.667|2|4|
> |RL|Full (4B)|4×H100|26|104|100|64|
>
> ECG QA
> |Stage|Trainable Params|GPUs|Wall-clock (h)|GPU-hours|Updates/Epochs|Batch Size|
> |---|---|---|---|---|---|---|
> |SFT|LoRA (16.5 M)|1×H100|1.35|1.35|3|4|
> |RL|Full (4B)|4×H100|27|108|100|64|
>
> ## Inclusion of Limitation Section
> Regarding the reviewer’s comment that “the paper does not explicitly discuss the limitations of the proposed approach,” the manuscript already includes an explicit Limitations subsection in Section 5, where we discuss computational trade-offs and scalability. We will expand the subsection in the revision to make these limitations clearer.
>
> **References**: [1]: Dau et al., “The UCR time series archive”, IEEE/CAA Journal of Automatica Sinica, 2019. [2] Haoxin et al., “A picture is worth a thousand numbers: Enabling llms reason about time series via visualization.”, Proceedings of the 2025 Conference of the Nations of the Americas Chapter of the Association for Computational Linguistics: Human Language Technologies, 2025 [3] Middlehurst et al., “Bake off redux: a review and experimental evaluation of recent time series classification algorithms.” arXiv preprint arXiv:2304.13029, 2023. [4] Godahewa et al., “Monash time series forecasting archive”., Neural Information Processing Systems Track on Datasets and Benchmarks, 2021. [5] Nie et al., “A time series is worth 64 words: Long-term forecasting with transformers”, International Conference on Learning Representations, 2023

---

> > ### Author Rebuttal · Reviewer_WcoD · 2026-04-04
> >
> > We thank the authors for the additional clarifications and for providing further experimental details. The ablation results help improve the understanding of the contribution of each component, and the discussion on computational cost is appreciated.
> >
> > However, my main concern regarding scalability remains only partially addressed. While the current experiments include moderately long sequences, the rebuttal does not provide sufficient evidence or analysis to demonstrate how the proposed approach behaves as sequence length or temporal resolution increases further. In particular, it remains unclear whether the key benefit of question-adaptive selective segment usage is maintained under such settings, and how the associated computational cost evolves.

---

> > > ### Author Response · Authors · 2026-04-05
> > >
> > > We thank the reviewer for this constructive feedback. We conducted additional experiments to directly evaluate ARTIST's scalability as sequence length increases.
> > >
> > > **Experimental setup.**
> > >
> > > Due to the lack of time series reasoning datasets with very long time series, we examine ARTIST’s scalability behavior by constructing extended-length variants of our longest dataset, RCW (original length: 4,000). For each test sample, we extended the time series to 8,000 and 12,000 by appending noisy copies of the original signal beyond the first 4,000 steps. The original [0, 4000) region remains unmodified and contains all task-relevant information, while the appended region consists of tiled copies with small additive Gaussian noise. The model was not re-trained; thus, this tests whether ARTIST's controller can generalize its segment selection to longer sequences at inference time. All results are averaged over 8 independent runs per sample.
> > >
> > > **Results.**
> > > |TS Length|Acc|F1|Time/example (min)|
> > > |---|---|---|---|
> > > |4000|77.00|50.00|1.88|
> > > |8000|75.66|51.21|1.895|
> > > |12000|76.11|48.37|1.91|
> > >
> > >
> > > Performance remains stable across all lengths, with less than 1.5 percentage points variation in accuracy from 4K to 12K, demonstrating that the controller successfully identifies the task-relevant temporal region even within sequences 3x longer than those seen during training. Equally important, per-example inference time increases by only 1.6% when tripling the sequence length, since ARTIST's computational cost is dominated by the controller-reasoner interaction over selected segments rather than encoding the time series.
> > > ARTIST’s scalability demonstrates a major advantage of disentanglement between the controller and reasoner in our hierarchical optimization approach. The reasoner’s behavior is optimized only across one interaction iteration and only receives the controller-selected segments rather than the full time series; thus, increasing the time series length should not affect the reasoner’s behavior. Meanwhile, the controller’s behavior is optimized over multiple interaction iterations to select segments one at a time until it deems that the reasoner has enough information to confidently solve the problem, at which it stops providing further segments. Since the number of task-relevant regions in a time series is generally determined by the nature of the question rather than the total sequence length, extending the time series does not necessarily increase the number of segments the controller needs to select; thus, the number of interaction iterations is not expected to vary greatly with increased time series length.

---

### Official Review · Reviewer_ifFu · 2026-03-13

**Soundness:** 2
**Presentation:** 1
**Significance:** 2
**Originality:** 1
**Overall Recommendation:** 4
**Confidence:** 3

**Summary:**

The paper proposes ARTIST, a framework designed for time series question-answering that utilizes reasoning in natural language with adaptive temporal segment selection. One LLM policy plays two roles through promptsL a controller to select the segment of the time series and to decide when to stop, and a reasoner that produces intermediate reasoning traces and answers based on the accumulated history. The training is also in two stages, where the model is fine-tuned in a supervised manner on reasoning traces, and then fine-tuned with reinforcement learning through self-play with a hierarchical optimization scheme. The controller is optimized for reliability and the reasoner is trained for correctness. Experiments cover six benchmarks in various domains, and ARIST improves average accuracy and uses less of the input series.

**Compliance With Llm Reviewing Policy:**

Affirmed.

**Final Justification:**

Given that the revision will contain the rebuttal, it is an interesting finding with reasonable explanations.

**Key Questions For Authors:**

- Please provide the detailed training cost for the RL stage and compare it with the SFT stage, along with the necessary number of rollout steps for convergence.
- How sensitive is the performance compared to L? A plot would be helpful.
- What would happen if the reasoner receives the entire time series but with the selected part highlighted?

**Limitations:**

yes

**Strengths And Weaknesses:**

### Strengths
- The core idea of focusing on segments of time-series and formulating it as a sequential problem is intuitive and distinct from prior work.
- Controller-reasoner design to decompose the tasks is also intuitive.
- The hierarchical RL training scheme seems to be a strong choice: measure consistency across multiple reasoner rollouts for more stable signal. Variance-guided sampling for the reasoner update is an efficient way to train the model without processing all rollouts.
- The experiments and ablation studies are thorough.

### Weaknesses
- When there is a ground truth data used to train the model in a supervised manner,
- The model struggles on Sleep-QA, but the explanation is not sufficient. The authors should investigate further whether the model struggles to learn EEG features, or whether it is the selector that is suffering for this data modality. A detailed study on both stages needs to be provided.
- The reasoning traces are generated with an oracle, and it is unclear if traces generated without oracle access would still help improve the performance.

---

> ### Author Rebuttal · Authors · 2026-03-31
>
> Thank you for your helpful feedback! We are glad that you found our formulation of time series reasoning and our adaptive model design to be intuitive and distinct from other works, and that you thought our choice of hierarchical policy optimization was strong and well-justified. We also appreciate your acknowledgement of our thorough benchmarking and ablation experiments.
>
> We have addressed your suggestions/concerns below and have also included the results of new experiments that we hope provide further insight and strengthen our paper. If you find your concerns addressed after reviewing our responses, we would greatly appreciate it if you considered raising your score. If there are any remaining questions/comments, please let us know and we would be happy to address them!
>
> ---
> ## W1 - Usage of SFT
> Regarding the concern over the use of SFT, our pipeline follows a now-common recipe in recent reasoning models: a supervised cold-start stage followed by RL. NeurIPS 2025 papers on hybrid and adaptive reasoning explicitly use this two-stage design, with SFT initializing the policy before RL refinement, rather than training purely from scratch (Qi et al., NeurIPS’25). This matters because RL for reasoning depends on the model having some probability of producing correct or partially correct trajectories at initialization; otherwise rewards are too sparse for effective optimization (Wang et al. NeurIPS’25, Yue et al., NeurIPS’25).
>
>
> ## W2 - Performance on Sleep-QA
> Sleep-QA is the most challenging dataset for all our baselines, not only for ARTIST (Table 1 in the paper). Following Table 2, which shows that VLMs perform better on Sleep-QA, we ran a new experiment to investigate whether this limitation is modality-driven, i.e., specific to tokenized time-series representations. We conducted an experiment where we replaced the backbone of ARTIST (Qwen3-4B) with a VLM (Qwen2.5-VL-3B) and represented the TS as plots instead of a tokenized sequence. Due to time constraints for this rebuttal, we performed SFT only. Table below shows that ARTIST achieves substantially higher performance under the vision-based representation, suggesting that the performance gap is primarily driven by the input modality rather than the segment selection or reasoning mechanism itself. TIME-MASTER (the strongest VLM on Sleep-QA) is included for reference.
>
> |Method|Acc|F1|
> |---|---|---|
> |ARTIST (VL)|**65**|**39.22**|
> |ARTIST |28.13|17.94|
> |TIME-MASTER|47.51|32.98|
>
> ## W3 - Oracle-generated reasoning traces for SFT
> The reviewer raises a concern about ARTIST's dependence on “oracle”-generated reasoning traces during SFT. To assess this, we conducted an experiment in which we injected noise into a subset of training reasoning traces. Specifically, we introduce noise by (i) replacing time-series descriptors with their semantic opposites, and (ii) injecting hallucinated observations by randomly inserting fabricated statements into the reasoning traces. We then perform SFT on ETI using these corrupted traces and report results averaged over 8 inference runs. The results show that ARTIST’s performance does not critically depend on having only the highest quality SFT reasoning traces from a frontier “oracle” model like GPT-5.
>
> |Noise %|Acc|F1|
> |---|---|---|
> |0%|85.12|85.11|
> |6%|84.5|84.16|
> |12%|83|85.5|
> |25%|85.6|85.3|
> |30%|85.06|85.08|
>
> ## Q1 - Training Cost of RL vs. SFT
> We have provided details on this in our response to W3 for Reviewer WcoD.
>
> ## Q2 - Sensitivity to the Max Interactions
> The reviewer asks about the sensitivity of ARTIST to the maximum number of controller-reasoner interaction rounds L. To address this, we conduct an ablation study varying L.
>
> |L|Acc|F1|
> |---|---|---|
> |1|84|84.62|
> |2|86.5|85.6|
> |3|87.03|87.1|
>
> ## Q3 -  Full TS with Highlighted Segments vs. Segment Selection
> The reviewer asks what would happen if the reasoner received the entire TS with the controller-selected regions highlighted, rather than only the selected segments. We ran experiments comparing these two settings. The results below show that our method performs better than providing the full TS with highlighted regions.
>
> We can empirically explain this: 1) When the reasoner is provided with a full TS with unhelpful segments highlighted, it can still achieve strong performance by just ignoring them and using other parts of the TS. Since controller’s learning signal comes from the reliability reward, this can falsely indicate to the controller that the highlighted segments are helpful, hampering the controller's ability to improve. 2) If the controller stops being able to learn how to provide helpful segments, the reasoner will be further incentivized to ignore the highlighted segments, and this would devolve into more naive methods that involve a fixed encoding of the entire TS.
>
> **ETI:**
> |Method|Acc|F1|
> |---|---|---|
> |ARTIST|87.03|87.10|
> |Highlight Segments|85.56|85.52|
>
> **TRQA:**
> |Method|Acc|F1|
> |---|---|---|
> |ARTIST|83.06|78.02|
> |Highlight Segments|74.84|72.95|

---

> > ### Author Rebuttal · Reviewer_ifFu · 2026-04-06
> >
> > The rebuttal addresses my concerns and the new results should be incorporated in the draft. I raise the score accordingly.

---

### Decision · Program_Chairs · 2026-04-30

**Decision:**

Accept (regular)

**Comment:**

This paper studies an important and increasingly relevant problem: answering natural-language questions over time series when the relevant evidence may be sparse, local, and question-dependent. The core idea of treating time-series reasoning as a sequential decision problem with adaptive segment selection is both intuitive and meaningful. Rather than forcing all information through a fixed full-series encoding, ARTIST explicitly separates where to look from how to reason, using a controller–reasoner architecture and hierarchical optimization. I find this to be a substantive contribution, and the reviewers generally agreed that the formulation is well motivated and technically coherent.

A major strength of the paper is the breadth of the empirical study. The method is evaluated on six benchmarks spanning multiple domains, with comparisons against strong LLM, VLM, and prior time-series reasoning baselines. The reported gains are substantial, and the paper also shows that ARTIST uses a smaller fraction of the input series while improving accuracy. Several reviewers further highlighted the quality of the ablations and the conceptual clarity of the controller–reasoner decomposition.

The main concerns raised in review centered on three issues: whether the gains were sufficiently disentangled across components, whether the approach scales to longer sequences, and whether some aspects of the framing and evaluation needed clarification. In my judgment, the rebuttal addressed these concerns credibly. The authors added clearer component-wise ablations, additional training and inference cost analysis, a Sleep-QA study isolating the role of modality, sensitivity analyses for interaction rounds and rollout parameters, and a longer-sequence scalability experiment on extended RCW variants. They also clarified the relation to dynamic visual search methods, corrected imprecise framing around hierarchical RL, and made explicit that the strongest baselines were fine-tuned on the same supervised data. These additions substantially strengthened the submission.

The paper is not without limitations. The current setting is restricted to univariate time series, and the strongest scalability evidence still comes from a synthetic extension of the longest benchmark rather than from a naturally occurring long-horizon dataset. In addition, one reviewer remained somewhat unconvinced by the evaluation protocol used in the new comparison to XAI baselines, particularly regarding whether all selected segments or only final-step segments should be used in the explanation comparison. I view these as reasonable caveats, but not as issues that undermine the main contribution.

Overall, I find this to be a solid paper with a clear idea, strong empirical support, and a rebuttal that materially improved the work. The adaptive segment-selection paradigm is likely to be of interest beyond the specific benchmarks studied here, and the paper opens a useful direction for question-adaptive reasoning over temporal data. I therefore recommend Weak Accept.